# A selective LIS1 requirement for mitotic spindle assembly discriminates distinct T-cell division mechanisms within the T-cell lineage

Jérémy Argenty, Nelly Rouquié, Cyrielle Bories, Suzanne Mélique, Valérie Duplan-Eche, Abdelhadi Saoudi, Nicolas Fazilleau, Renaud Lesourne*

Toulouse Institute for Infectious and Inflammatory Diseases (Infinity), INSERM UMR1291, CNRS UMR5051, University Toulouse III, Toulouse, France

**Abstract** The ability to proliferate is a common feature of most T-cell populations. However, proliferation follows different cell-cycle dynamics and is coupled to different functional outcomes according to T-cell subsets. Whether the mitotic machineries supporting these qualitatively distinct proliferative responses are identical remains unknown. Here, we show that disruption of the microtubule-associated protein LIS1 in mouse models leads to proliferative defects associated with a blockade of T-cell development after β-selection and of peripheral CD4+ T-cell expansion after antigen priming. In contrast, cell divisions in CD8+ T cells occurred independently of LIS1 following T-cell antigen receptor stimulation, although LIS1 was required for proliferation elicited by pharmacological activation. In thymocytes and CD4+ T cells, LIS1 deficiency did not affect signaling events leading to activation but led to an interruption of proliferation after the initial round of division and to p53-induced cell death. Proliferative defects resulted from a mitotic failure, characterized by the presence of extra-centrosomes and the formation of multipolar spindles, causing abnormal chromosomes congression during metaphase and separation during telophase. LIS1 was required to stabilize dynein/dynactin complexes, which promote chromosome attachment to mitotic spindles and ensure centrosome integrity. Together, these results suggest that proliferative responses are supported by distinct mitotic machineries across T-cell subsets.

*For correspondence:
renaud.lesourne@inserm.fr

Competing interest: The authors declare that no competing interests exist.

## Editor's evaluation

This paper reports a fundamental finding that the requirement for the dynein binding protein, Lis1, is differentially required by different T cell lineages. The paper provides compelling evidence that the regulation of cell division and subsequent fate differs across T cell lineages. This work will be of interest to researchers focussed on T cell biology and mitosis.

## Introduction

Proliferation enables the expansion, differentiation, and maintenance of T cells at different stages of their life cycle. It is required for the rapid growth of antigen-specific T cells, which is important for efficient control of infection. In this context, the initiation of cell division is primarily driven by signals triggered by the T-cell antigen receptor (TCR), which recognizes self or foreign peptides bound to the major histocompatibility complex (pMHC) at the surface of antigen-presenting cells (APCs). Proliferation is also important during T-cell development as it enables the expansion of immature CD4-CD8-thymocytes (referred to as double-negative [DN] thymocytes) that have successfully rearranged the

TCR β-chain and their differentiation into CD4+CD8+thymocytes (referred to as double-positive [DP] thymocytes) (*Kreslavsky et al., 2012*; *Pénit et al., 1995*). At these stages, proliferation is mainly driven by coordinated signaling events triggered by the pre-TCR and by the Notch receptor (*Ciofani et al., 2004*; *Maillard et al., 2006*). Slow proliferative events are also induced in peripheral T cells to maintain a functional and diversified pool of lymphocytes. Such homeostatic proliferation is triggered in response to TCR stimulation by self-pMHC ligands and to specific cytokines (*Sprent and Surh, 2011*).

CD4+ T helper cells and CD8+ cytotoxic T cells harbor different proliferative characteristics in response to TCR stimulation. CD4+ T cells require repeated TCR stimulation to efficiently divide and show a relatively restricted expansion rate following antigen priming, while CD8+ T cells divide rapidly after single TCR stimulation (*Foulds et al., 2002*; *Seder and Ahmed, 2003*). Cell division is associated to the acquisition of effector function in CD8+ T cells (*Chang et al., 2007*; *Arsenio et al., 2014*). The fate decision between the effector and memory lineages in CD8+ T cells has been proposed to occur as early as the first round of division through asymmetric divisions (*Arsenio et al., 2014*), which enables the unequal partitioning of cell fate determinants in daughter cells (*Chang et al., 2007*). The role of cell division in the acquisition of CD4+ T cells effector function has been controversial (*Bird et al., 1998*; *Ben-Sasson et al., 2001*). Asymmetric divisions were also reported in CD4+ T cells (*Chang et al., 2011*; *Nish et al., 2017*), but the contribution of such processes to T helper lineage diversification, which primarily depends on cytokine stimuli, remains also debated (*Cobbold et al., 2018*). Together, these findings suggest that different cell division dynamics and organization might govern proliferation in CD4+ and CD8+ T cells to ensure different functional outcomes. Whether the mitotic machinery supporting these qualitatively distinct proliferative responses are identical is unknown.

Lissencephaly gene 1 (LIS1, also known as PAFAHB1) is a dynein-binding protein that has important function during brain development (*Markus et al., 2020*). LIS1 is involved in the proliferation and migration of neural and hematopoietic stem cells (*Yingling et al., 2008*; *Reiner and Sapir, 2013*; *Zimdahl et al., 2014*). It binds to the motor protein complex dynein and regulates the dynamic of its interaction with microtubules (*Markus et al., 2020*; *Yamada et al., 2008*; *Huang et al., 2012*), as well as its ability to form active complex with the multimeric protein complex dynactin (*Htet et al., 2020*; *Elshenawy et al., 2020*; *Wang et al., 2013*). Those complexes are required for the long transport of cargos toward the minus end of microtubules (*McKenney et al., 2014*; *Ayloo et al., 2014*; *Urnavicius et al., 2015*; *Schlager et al., 2014*) and are important for a wide variety of cellular processes, including the accumulation of γ-tubulin at the centrosome (*Young et al., 2000*; *Blagden and Glover, 2003*) and the efficient formation of mitotic spindle poles during metaphases (*Quintyne and Schroer, 2002*). Recently, we identified LIS1 as a binding partner of the T-cell signaling protein THEMIS (*Zvezdova et al., 2016*; *Garreau et al., 2017*), which is important for thymocyte positive selection, suggesting that LIS1 could exhibit signaling function during T-cell development. LIS1 is required in several cellular models for chromosome congression and segregation during mitosis and for the establishment of mitotic spindle pole integrity (*Moon et al., 2014*). However, the impact of LIS1 deficiency on cell division varies according to cell types and stimulatory contexts. For example, LIS1 is essential to symmetric division of neuroepithelial stem cells prior neurogenesis, whereas LIS1 deficiency has a moderate impact on asymmetric division associated to the differentiation neuroepithelial stem cells in neural progenitors (*Yingling et al., 2008*). Previous studies also suggest that LIS1 is dispensable for the expansion of CD8+ T cells induced following antigen priming (*Ngoi et al., 2016*). Together, these findings suggest that LIS1 could have stage- or subset-specific effects on T-cell mitosis, which might discriminate distinct cellular outcomes.

Here, we selected LIS1 as a candidate molecule to explore whether T-cell proliferative responses could be supported by distinct mitotic machineries across different T-cell subsets, such as immature thymocytes as well as CD4+ and CD8+ T cells. Using different *Cre*-inducible models, we identified a selective LIS1 requirement for mitosis in thymocytes and peripheral CD4+ T cells following β-selection and antigen priming, respectively. In contrast, the disruption of LIS1 had little impact on CD8+ T cell proliferation mediated by the TCR. In thymocytes and CD4+ T cells, LIS1 deficiency led to a disruption of dynein–dynactin complexes, which was associated with a loss of centrosome integrity and with the formation of multipolar spindles. These mitotic abnormalities conducted to abnormal chromosomes congression and separation during metaphase and telophase, and to aneuploidy and

p53 upregulation upon cell division. Together, our results suggest that the mechanisms that support mitosis within the T-cell lineage could vary across T-cell subsets according to the functional outcomes to which they are coupled.

## Results

### Lis1 deficiency leads to an early block of T- and B-cell development

To evaluate the role of LIS1 during T-cell development, we conditionally disrupt *Pafah1b1*, the gene encoding LIS1, using a Cre recombinase transgene driven by the human *cd2* promoter, which is upregulated in T- and B-cell progenitors (*Greaves et al., 1989*) (mouse line referred to hereafter as CD2-Lis1 cKO). As a control, *Pafah1b1*flox/flox mice (referred to as control mice) were used. Analysis of CD4 and CD8 surface staining in the thymus shows that the loss of LIS1 in the *Cd2-Cre* model leads to a major block of thymocyte maturation at the transition from the DN stage to the DP stage, which is associated with a strong decrease in DP, CD4, and CD8 single-positive (SP) thymocytes numbers but normal numbers of DN thymocytes (*Figure 1A*). Numbers of peripheral CD4+ and CD8+ T cells were also dramatically decreased in CD2-Lis1 cKO mice compared to that in control mice (*Figure 1—figure supplement 1A*). Analysis of CD25 and CD44 surface staining on DN thymocytes showed that the numbers of DN4 (CD25-CD44-) thymocytes were strongly decreased in LIS1-deficient mice, whereas the numbers of DN3 (CD25+CD44-) and DN2 (CD25+CD44+) thymocytes were increased, pointing out a defect at the transition from the DN3 to the DN4 stages (*Figure 1B*). The percentages and numbers of thymocyte in each subset were comparable to control mice in *Cd2-cre Pafah1b1*flox/+ mice (referred to as CD2-Lis1 cKO-het), indicating that LIS1 hemi-zygote expression is sufficient to promote T-cell development (*Figure 1—figure supplement 1B*). Lower numbers of B cells were also detected in LIS1-deficient mice (*Figure 1—figure supplement 1A*). Analysis of B-cell development in the bone marrow indicates a strong decrease in the numbers of B220+CD19+ pro-B (IgM-c-kit+), pre-B (IgM-c-kit-), and immature B cells (IgM+c-kit-), whereas numbers of pre-pro-B cells (B220+CD19-) were normal, suggesting a defect of maturation of pre-pro-B cells into pro-B cells (*Figure 1—figure supplement 1C*). Together, these data indicate that LIS1 is essential for early stages of T- and B-cell development.

### LIS1 is required for thymocyte proliferation after the β-selection checkpoint

One critical developmental step at the DN3 to DN4 transition is the formation of a functional TCR β chain, which associates with the pTα chain upon successful rearrangement to form the pre-TCR. Pre-TCR formation triggers signaling events, which lead to the upregulation of CD5 and, together with Notch and the IL-7 receptor (IL-7R) stimulation, to the initiation of several division cycles and to further maturation of thymocytes into DN4 thymocytes (*Kreslavsky et al., 2012*; *Pénit et al., 1995*; *Maillard et al., 2006*; *Boudil et al., 2015*; *Azzam et al., 1998*). The percentages of DN3 thymocytes that express the TCRβ chain and CD27, a cell surface maker of the β-selection checkpoint (*Taghon et al., 2006*), were lower in CD2-Lis1 cKO mice compared with those in control mice expressing LIS1, suggesting that LIS1 might be important for the rearrangement of the TCRβ chain and/or for the expansion of cells that successfully rearranged the TCRβ chain (*Figure 1C*). The expression level of CD5 was slightly increased in CD2-Lis1 cKO DN3 thymocytes compared with that in control DN3 cells, whereas IL-7R cell surface levels were not affected by LIS1 expression, suggesting that LIS1 was not required for functional pre-TCR assembly but rather for the expansion of DN3 thymocytes after the β-selection checkpoint (*Figure 1D and E*). Notch signaling leads to increased cell sizes of thymocyte after β-selection and to the upregulation of the transferrin receptor CD71 (*Ciofani and Zúñiga-Pflücker, 2005*; *Kelly et al., 2007*). The loss of LIS1 did not affect these two parameters, suggesting that LIS1 is dispensable for Notch-mediated signaling (*Figure 1E*). To evaluate whether LIS1 is important for the proliferation of DN3 thymocytes following the β-selection checkpoint, we quantified DN cells that have duplicated DNA copies prior and after the β-selection checkpoint. Thymocytes with duplicated DNA copies could not be detected prior the β-selection checkpoint in wild-type and LIS1-deficient mice (*Figure 1F*). Approximately 10% of thymocytes were in the G2/M phase of cell cycle after β-selection in wild-type mice, whereas this proportion rose to 20% in LIS1-deficient mice, suggesting a possible failure of LIS1-deficient thymocytes to successfully complete division cycles (*Figure 1F*).

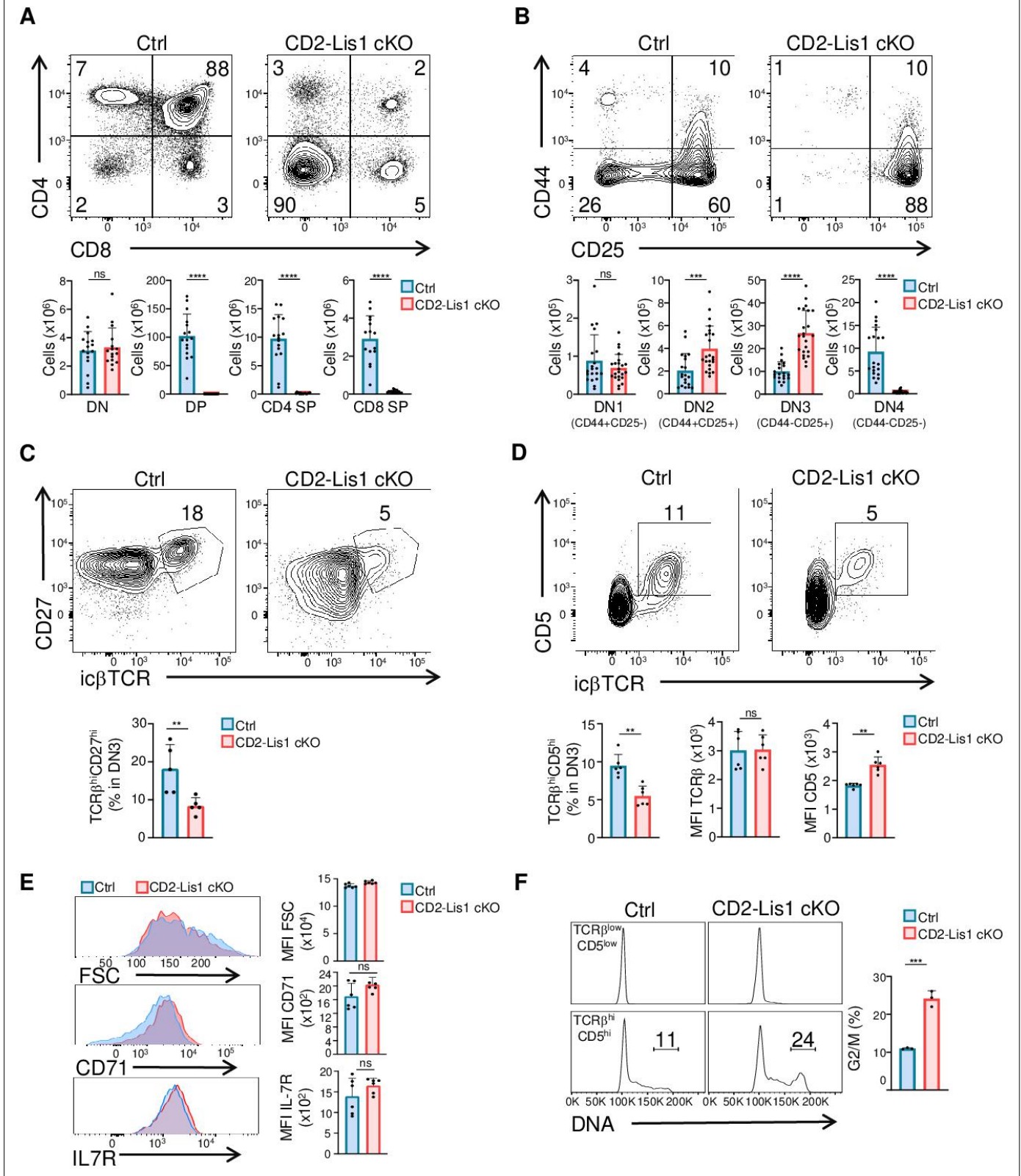

**Figure 1.** LIS1 is required for T-cell development following the β-selection checkpoint. Phenotypic analyses of thymocytes from control and CD2-Lis1 cKO mice. (**A**) Dot plots show CD4 versus CD8 surface staining on thymocytes from control and CD2-Lis1 cKO mice. Histogram bars represent the numbers of thymocytes in each indicated subset from mice of the indicated genotype. Data are mean ± SD and represent a pool of four independent experiments each including n = 3–4 mice per group. (**B**) Dot plots show CD44 versus CD25 surface staining on CD4-CD8- [DN] thymocytes from control

*Figure 1 continued on next page*

*Figure 1 continued*

and CD2-Lis1 cKO mice. Histogram bars represent the numbers of thymocytes in each indicated subset from mice of the indicated genotype. Data are mean ± SD and represent a pool of five independent experiments each including n = 4–5 mice per group. (**C**) Dot plots show CD27 versus TCRβ intracellular staining on DN3 thymocytes from control and CD2-Lis1 cKO mice. Histogram bars represent the percentages of TCRβ$^{hi}$CD27$^{hi}$ thymocytes in DN3 thymocytes. Data are mean ± SD and represent a pool of two independent experiments each including n = 2–3 mice per group. (**D**) Dot plots show CD5 versus TCRβ intracellular staining on DN3 thymocytes from control and CD2-Lis1 cKO mice. Histogram bars represent the percentages of TCRβ$^{hi}$CD5$^{hi}$ thymocytes in DN3 thymocytes and the MFI of TCRβ and CD5 in DN3 TCRβ$^{hi}$CD5$^{hi}$ thymocytes from mice of the indicated genotype. Data are mean ± SD and represent a pool of two independent experiments each including n = 3 mice per group. (**E**) Histogram graphs show IL-7R, CD71 surface staining and forward-scatter (FSC) on DN3 thymocytes expressing the TCRβ chain. Histogram bars represent the MFI of IL-7R, CD71, and FSC in the indicated DN3 thymocytes subsets. Data are mean ± SD and represent a pool of two independent experiments each including n = 3 mice per group. (**F**) Histogram graphs show DNA intracellular staining on DN3 thymocytes from the indicated subsets. The percentages represent cells in the G2/M phase of cell cycle. Histogram bars represent the percentages of DN3 TCRβ$^{hi}$CD5$^{hi}$ thymocytes in the G2/M phase of cell cycle. Data are mean ± SD and represent a pool of three independent experiments each including n = 1 mouse per group. Unpaired two-tailed Mann–Whitney *t* tests were performed for all analyses. \*\*p<0.01; \*\*\*p<0.001; \*\*\*\*p<0.0001.

The online version of this article includes the following source data and figure supplement(s) for figure 1:

**Source data 1.** LIS1 is required for T-cell development following the β-selection checkpoint.

**Figure supplement 1.** LIS1 is required for B-cell development and is effective at single-gene dosage during T-cell development.

**Figure supplement 1—source data 1.** LIS1 is required for B-cell development and is effective at single-gene dosage during T-cell development.

To directly address this hypothesis, we analyzed the proliferation of DN3 thymocytes upon stimulation with OP9-Dl1 cells, a bone marrow-derived stromal cell line that ectopically expresses the Notch ligand, Delta-like 1 (Dl1), and which induces efficient T-cell lymphopoiesis from the DN stages to the DP stage (*Schmitt et al., 2004*). We observed that the percentages of cells that proliferate in response to OP9-Dl1 stimulation were strongly decreased in the absence of LIS1 (*Figure 2A*). This was associated with a failure of DN3 cells to effectively differentiate into CD25-CD44- DN4 cells (*Figure 2B*). The TCRβ chain and the receptor CD5 were upregulated normally after stimulation, indicating that the defect in proliferation was not the consequence of defects in early stimulatory signals required for proliferation and differentiation (*Figure 2C*). The loss of LIS1 also did not affect the expression of CD71 (*Figure 2D*) and Bcl-2 (*Figure 2E*), which depends on Notch and IL-7R signaling, respectively (*Akashi et al., 1997*; *Maraskovsky et al., 1997*), suggesting that LIS1 does not operate downstream of these receptors. By contrast, cell cycle analysis showed that the loss of LIS1 led to a strong accumulation of cells at the G2/M stage, indicative of ineffective division processes after the DNA duplication phase (*Figure 2F*). Together, those results suggest that LIS1 controls cellular events that are required for the efficient division of thymocytes after the β-selection checkpoint.

## LIS1 is required for TCR-mediated proliferation in CD4+ T cells

Previous studies suggested that LIS1-deficient CD4+ and CD8+ T cells fail to proliferate in response to cytokine-driven homeostatic signals but successfully divide in response to TCR cross-linking in vitro or following infection with a *Listeria monocytogenes* strain expressing ovalbumin (*Ngoi et al., 2016*). Since the loss of LIS1 had such a strong impact on thymocyte proliferation following pre-TCR stimulation, we decided to compare the role of LIS1 in the proliferation of CD4+ and CD8+ T cells in response to TCR engagement.

To examine the role of LIS1 in peripheral T cells, we conditionally disrupt *Pafah1b1* using a Cre recombinase transgene driven by the *Cd4* promoter, which is upregulated at the DP stage after the proliferation step of DN3-DN4 thymocytes (mouse line referred to hereafter as CD4-Lis1 cKO). We observed that the loss of LIS1 at this stage of development did not affect the percentages and numbers of DN, DP, and SP thymocytes (*Figure 3—figure supplement 1A*). Normal numbers of mature TCR$^{hi}$CD24$^{low}$ SP thymocytes were also generated in the absence of LIS1 (*Figure 3—figure supplement 1B*). The maturation of DP thymocytes into TCR$^{hi}$CD4 SP thymocytes occurred also normally in CD4-Lis1 cKO mice expressing a fixed MHC class II–restricted αβ-TCR transgene (AND), suggesting that LIS1 is not essential for positive selection (*Figure 3—figure supplement 1C*). As previously reported in a similar conditional knockout model (*Ngoi et al., 2016*), the deletion of LIS1 led to a dramatic decrease in peripheral CD4+ and CD8+ T cells numbers (*Figure 3—figure supplement 1D*). This defect was previously imputed to a reduced ability of CD4+ and CD8+ T cells to proliferate in response to cytokine-driven homeostatic signals (*Ngoi et al., 2016*). In contrast, the

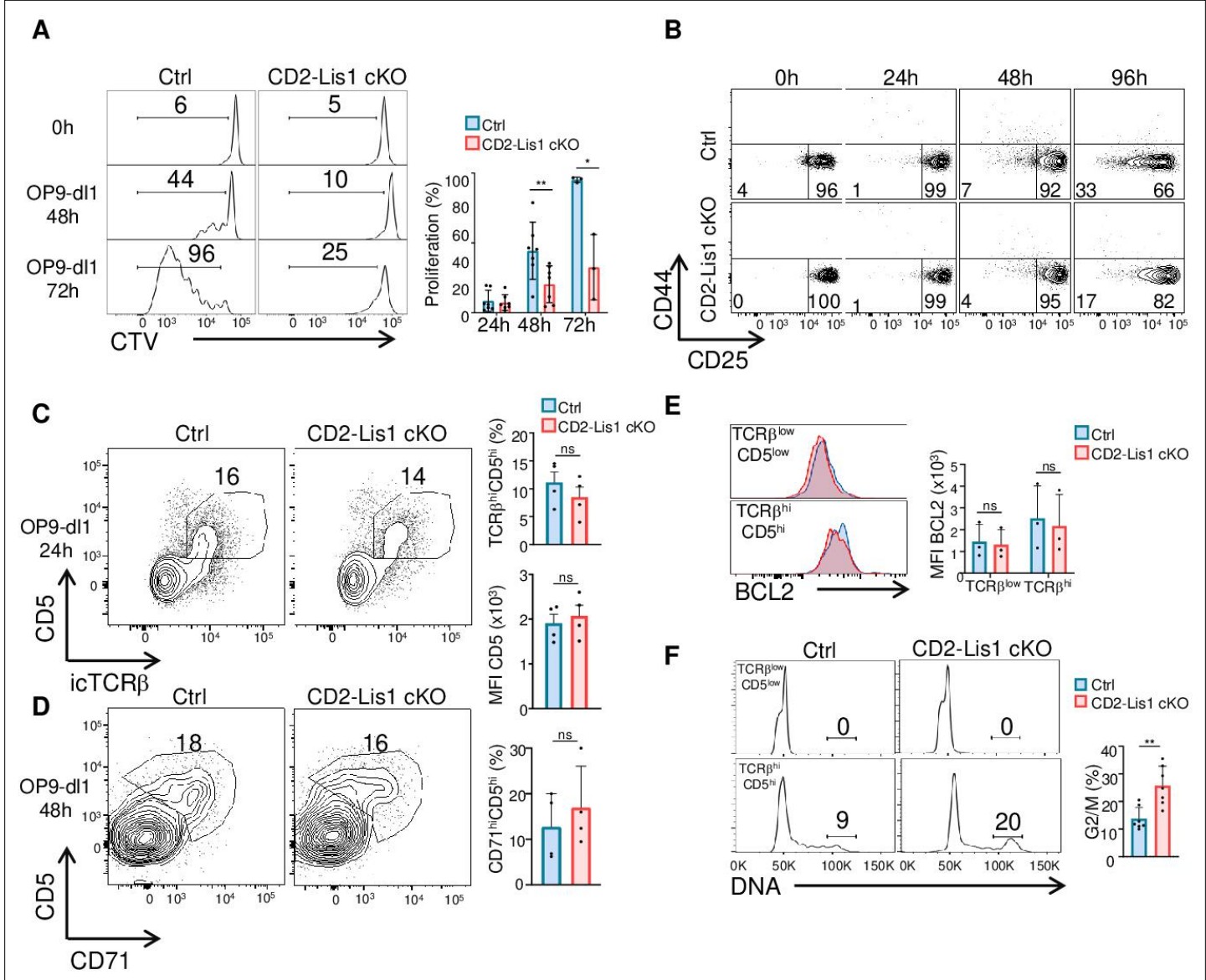

**Figure 2.** LIS1 is required for the proliferation of immature thymocytes after the β-selection checkpoint. (**A**) CD5lo DN3 thymocytes from control and CD2-Lis1 cKO mice were stained with CellTrace violet (CTV) and stimulated with OP9-Dl1 cells for 48 or 72 hr. The histogram graph shows CTV dilution. Bar graphs represent the proliferation of cells determined by flow cytometry at 24, 48, and 72 hr after stimulation. Data are mean ± SD and represent 3–7 independent experiments each including n = 1–2 pooled mice per group. (**B**) CD5lo DN3 thymocytes from control and CD2-Lis1 cKO mice were stimulated with OP9-Dl1 cells for the indicated periods of time. Dot plots show CD44 versus CD25 surface staining on thymocytes from control and CD2-Lis1 cKO mice. Data are representative of three independent experiments each including n = 1–2 pooled mouse per group. (**C**) CD71lo DN3 thymocytes from control and CD2-Lis1 cKO were stimulated with OP9-Dl1 cells for 24 hr. Dot plots show CD5 versus TCRβ intracellular staining on thymocytes. Histogram bars represent the percentages of TCRβhiCD5hi thymocytes in DN3 thymocytes and the MFI CD5 in DN3 TCRβhiCD5hi thymocytes from mice of the indicated genotype. Data are mean ± SD and represent four independent experiments each including n = 1–2 pooled mice per group. (**D**) CD71lo DN3 thymocytes from control and CD2-Lis1 cKO mice were stimulated with OP9-Dl1 cells for 48 hr. Dot plots show CD5 versus CD71 staining on CTVhi thymocytes. Histogram bars represent the percentages of CD71hiCD5hi thymocytes in CTVlo DN3 thymocytes. Data are mean ± SD and represent four independent experiments each including n = 1–2 pooled mice per group. (**E**) CD71lo DN3 thymocytes from control and CD2-Lis1 cKO mice were stimulated with OP9-Dl1 cells for 24 hr. The histogram graph shows BCL-2 intracytoplasmic staining in TCRβloCD5lo and TCRβhiCD5hi thymocyte subsets. Histogram bars represent the MFI of BCL-2 in the indicated DN3 thymocyte subsets. Data are mean ± SD and represent three independent experiments each including n = 1–2 pooled mice per group. (**F**) CD71lo DN3 thymocytes from control and CD2-Lis1 cKO mice were stimulated with OP9-Dl1 cells for 48 hr. Histogram graphs show DNA intracellular staining on thymocytes from the indicated DN3 subsets. The indicated percentages represent cells in the G2/M phase of cell cycle. Histogram bars represent the percentages of DN3 TCRβhiCD5hi thymocytes in the G2/M phase of cell cycle. Data are mean ± SD and represent six independent experiments each including n = 1–2 pooled mice per group. (**A**) Unpaired two-tailed Welch t tests were performed. (**C–E**) Unpaired two-tailed Mann–Whitney t tests were performed. *p<0.05, **p<0.01.

*Figure 2 continued on next page*

*Figure 2 continued*

The online version of this article includes the following source data for figure 2:

**Source data 1.** LIS1 is required for the proliferation of immature thymocytes after the β-selection checkpoint.

mono-allelic deletion of LIS1 in the *Cd4-cre* model (referred to as CD4-Lis1 cKO-het) did not affect the numbers of CD4+ and CD8+ T cells (*Figure 3—figure supplement 1E*), suggesting that reduced LIS1 dosage does not affect T-cell homeostasis.

We next evaluated the effect of LIS1 deficiency on the proliferation of CD4+ and CD8+ T cells following TCR stimulation. We observed that the percentages of proliferating CD4+ T cells were strongly decreased in the absence of LIS1 following TCR stimulation (*Figure 3A*). The analysis of cell percentages in each division cycle showed that LIS1-deficient CD4+ T cells successfully performed the first cycle of division but failed to divide further and accumulated at this stage (*Figure 3A*). Similar results were obtained following stimulation with phorbol 12-myristate 13-acetate (PMA) and ionomycin, indicating that LIS1-dependent effects on CD4+ T-cell proliferation were not dependent on proximal TCR signaling events (*Figure 3A*). In contrast, monoallelic deletion of the LIS1 encoding gene did not affect the rate of proliferating CD4+ T cells following TCR stimulation (*Figure 3—figure supplement 2A*). Activation markers such as CD25 and CD69 were also upregulated normally in the absence of LIS1, indicating that more distal TCR signaling events were not affected by LIS1 deficiency (*Figure 3B*). Cell-cycle analysis show that CD4+ T cells with duplicated DNA copies accumulated in LIS1-deficient T cells compared with that in control cells following stimulation (*Figure 3C*). Contrasting with the strong effect observed on CD4+ T-cell proliferation, the loss of LIS1 had a rather modest impact on the total fraction of CD8+ T cells that proliferate in response to TCR cross-linking and on the fraction of cells that had successfully divided after the first division cycle (*Figure 3D*). The loss of LIS1 also did not result in the accumulation of CD8+ T cells with duplicated DNA copies (*Figure 3C*). LIS1 was not detected in cell extracts from both CD4+ and CD8+ T cells from CD4-Lis1 cKO mice, indicating that the mild impact of LIS1 on CD8+ T-cell proliferation was not the consequence of the remaining expression of LIS1 in this subset (*Figure 3—figure supplement 2B*). Also, the stimulation of CD8+ T cells with PMA and ionomycin led to an important decrease in the total fraction of proliferating T cells, suggesting that cell divisions in CD8+ T cells are controlled by different mechanisms, which vary according to their LIS1 dependency based on the context of stimulation.

To determine whether LIS1 controls the proliferation of CD4+ T cells in response to antigen stimulation in vivo, we crossed CD4-Lis1 cKO mice with transgenic mice expressing the allotypic marker CD45.1 and the class-II restricted OT2 TCR specific for the chicken ovalbumin 323–339 peptide. CD4+ T cells from OT2+CD4-Lis1 cKO and control mice were stained with CellTrace violet (CTV) and injected into C57Bl/6 mice expressing the allotypic marker CD45.2+. Mice were next immunized with ovalbumin and CD45.1+CD4+T cells were analyzed in the spleen at days 2, 3, and 7 after immunization. The numbers of LIS1-deficient CD45.1+CD4 +T cells in the spleen were similar to those of control cells at day 2 after immunization, indicating that the loss of LIS1 did not affect the ability of CD4+ T cells to migrate into the spleen (*Figure 3E*). At this stage, the percentages of divided cells were very low and were not significantly different according to LIS1 expression. At day 3 after immunization, we observed a large fraction of divided control CD45.1+CD4+ T cells, with the majority of cells having completed more than two rounds of division (*Figure 3E and F*). By contrast, the fraction of divided cells was strongly decreased in the absence of LIS1 with almost a complete failure of those cells to engage more than one division cycle (*Figure 3E and F*). Numbers of LIS1-deficient CD4+CD45.1+ T cells were strongly decreased compared to control CD4+CD45.1+ T cells that express LIS1 (*Figure 3E*). Of note, the expression level of CD44 on undivided CD4+CD45.1+ T cells was similar whether or not LIS1 was expressed, suggesting that LIS1 was not required for CD4+ T-cell activation in vivo (*Figure 3F*). The loss of LIS1 also resulted in a marked decrease in the percentages and numbers of CD4+CD45.1+ T cells at day 7 after immunization (*Figure 3—figure supplement 2C*). Together, these results suggest that CD4+ and CD8+ T cells engage distinct cell division mechanisms upon antigen priming that diverge in their requirement for LIS1.

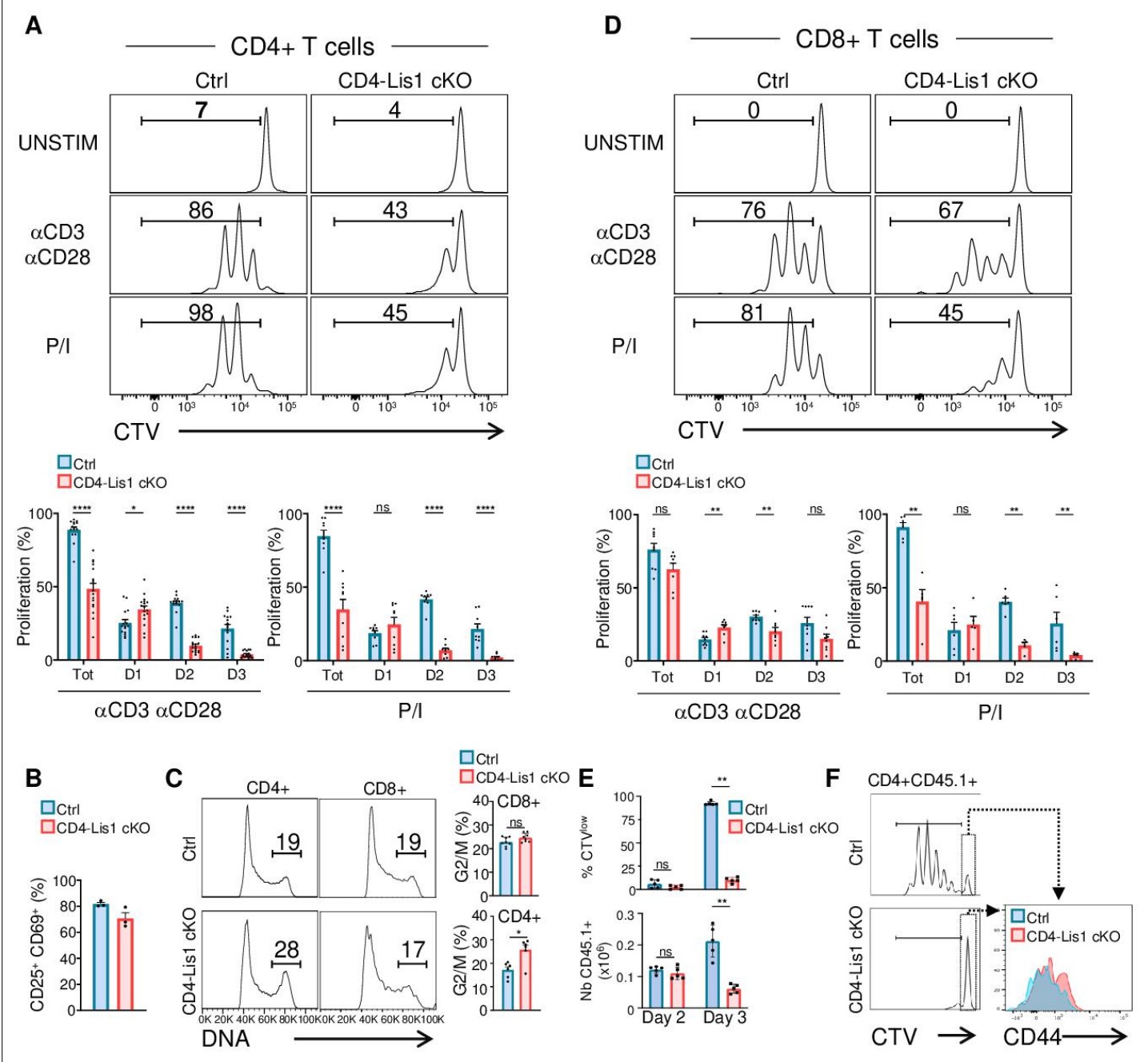

**Figure 3.** LIS1 is required for the proliferation of CD4+ T cells in response to antigen stimulation. (**A**) CD4+ T cells from control and CD4-Lis1 cKO mice were stained with CellTrace violet (CTV) and stimulated with anti-CD3 and anti-CD28 antibodies or with phorbol 12-myristate 13-acetate (PMA) and ionomycin (P/I) for 72 hr. The histogram graphs show CTV dilution. Bar graphs represent the percentages of cells that divided at least one-time (Tot.) or that divided one, two, or three times (D1, D2, D3) as determined by flow cytometry at 72 hr after stimulation. Data are mean ± SD and represent five independent experiments each including n = 3 mice per group. (**B**) CD4+ T cells from control and CD4-Lis1 cKO mice were stimulated with anti-CD3 and anti-CD28 antibodies for 24 hr. Bar graphs represent the percentages of cells expressing CD25 and CD69 as determined by flow cytometry. Data are mean ± SD and represent two independent experiments each including n = 1–2 mice per group. (**C**) CD4+ and CD8+ T cells from control and CD4-Lis1 cKO mice were stimulated with anti-CD3 and anti-CD28 antibodies for 48 hr. Histogram graphs show DNA intracellular staining on CD4+ and CD8+ T cells. The indicated percentages represent cells in the G2/M phase of cell cycle. Histogram bars represent the percentages of CD4+ and CD8+ T cells in the G2/M phase of cell cycle. Data are mean ± SD and represent two independent experiments each including n = 3 mice per group. (**D**) CD8+ T cells from control and CD4-Lis1 cKO mice were stained with CTV and stimulated with anti-CD3 and anti-CD28 antibodies or with PMA and ionomycin (P/I) for 72 hr. The histogram graph shows CTV dilution. Bar graphs represent the percentages of cells that divided at least one-time (Tot.) or that divided one, two, or three times (D1, D2, D3) as determined by flow cytometry at 72 hr after stimulation. (**E, F**) C57BL/6j mice (CD45.2+) were injected i.v. with

*Figure 3 continued on next page*

*Figure 3 continued*

CTV-stained CD45.1+CD4+ T cells from OT2 and OT2 CD4-Lis1 cKO mice. Mice were then immunized with ovalbumin emulsified in RIBI. Proliferation of CD45.1+CD4+T cells was analyzed at days 2 and 3 after immunization. (**E**) Bar graphs represent the proliferation and numbers of CD45.1+CD4+T cells as determined by flow cytometry at days 2 and 3 after immunization. Data are mean ± SD and are representative of one experiment out of two independent experiments each including n = 5 mice per group. (**F**) The histogram graph shows CTV dilution in CD45.1+CD4+T cells at day 3 after immunization. Histograms overlay shows CD44 surface staining on undivided CD45.1+CD4+T cells at day 3 after immunization. Data are representative of one experiment out of two independent experiments each including n = 5 mice per group. Unpaired two-tailed Mann–Whitney *t* tests were performed for all analyses. *p<0.05; **p<0.01; ***p<0.001; ****p<0.0001.

The online version of this article includes the following source data and figure supplement(s) for figure 3:

**Source data 1.** LIS1 is required for the proliferation of CD4+ T cells in response to antigen stimulation.

**Figure supplement 1.** Normal T-cell development in CD4-Lis1 cKO mice.

**Figure supplement 1—source data 1.** Normal T-cell development in CD4-Lis1 cKO mice.

**Figure supplement 2.** Effect of LIS1 haploid and diploid deficiency on CD4+ T-cell proliferation and expansion.

**Figure supplement 2—source data 1.** Effect of LIS1 haploid and diploid deficiency on CD4+ T-cell proliferation and expansion.

## LIS1-dependent control of chromosome alignment during metaphase is required for effective mitosis

We next aimed to more precisely characterize the role of LIS1 during the division of CD4+ T cells. Our data suggest a block either at the G2 or the M phase of cell cycle in LIS1-deficient thymocytes and CD4+ T cells (*Figure 1E*, *Figure 2E*, and *Figure 3C*). We used image stream flow cytometry to discriminate cells with duplicated DNA copies that contain chromosomes (in M phase) from cells that have uncondensed DNA (in G2 phase). CD4-Lis1 cKO and control CD4+ T cells were stimulated for 48 hr with anti-CD3 and anti-CD28 antibodies and stained with DAPI. Analysis was next performed on cells with duplicated DNA copies. The Bright Detail Intensity (BDI) feature on the DAPI channel, which evaluates areas of peak fluorescence intensity after subtraction of background fluorescence, was selected for its ability to automatically discriminate cells in M and G2 phases, as illustrated in *Figure 4A*. The percentages of mitotic CD4+ T cells were increased in the absence of LIS1, suggesting that LIS1-deficient CD4+ T cells fail to complete mitosis. To determine more precisely the stage of mitosis at which this defect occurs, we analyzed whether LIS1 was required for cells to successfully reach metaphase. CD4-Lis1 cKO and control CD4+ T cells were stimulated for 48 hr with anti-CD3 and anti-CD28 antibodies and synchronized with nocodazole for 18 hr prior treatment with MG132 to induce metaphase arrest. The percentages of cells in metaphase were evaluated by image stream flow cytometry using the 'Elongatedness' parameter, which calculates the length to width ratios (L/W) on predefined DAPI masks. CD4+ T cells with L/W ratios superior to 1.5 show aligned chromosomes patterns representative of metaphase (*Figure 4B*). This analysis showed that the percentages of cells that successfully reached metaphase were strongly reduced in the absence of LIS1 (*Figure 4B*). To more precisely characterize mitotic events that could be affected by LIS1 deficiency, we next analyzed the course of mitosis in CD4-Lis1 cKO and control CD4+ T cells by time-lapse microscopy. We observed that both CD4-Lis1 cKO and control CD4+ T cells successfully condensated their DNA to form chromosomes (*Figure 4C*, *Videos 1–3*). However, chromosomes remained disorganized in CD4-Lis1 cKO CD4+ T cells and failed to segregate rapidly after condensation compared with those in control cells (*Figure 4C*). At the final step of mitosis, LIS1-deficient CD4+ T cells either failed to divide (*Figure 4C and D*, *Video 2*) or divided with an apparent unequal repartition of chromosomes in daughter cells (*Figure 4C and D*, *Video 3*), which was associated with the formation of multiple nuclei or multilobed nuclei (*Figure 4C*, *Video 2*). Confirming the observations based on time-lapse microscopy, quantitative analysis on G2 cells selected by image stream showed that the percentages of cells with multiple nuclei were strongly increased in the absence of LIS1 (*Figure 4E*).

The abnormal repartition of chromosomes in daughter cells, so called aneuploidy, is generally associated with the upregulation of the tumor suppressor p53, which contributes to eliminate cells through apoptotic processes prior the emergence of possible oncogenic transformation (*Kastenhuber and Lowe, 2017*). To determine whether impaired mitosis associated to LIS1 deficiency leads to apoptosis, we analyzed the percentages of apoptotic cells in undivided and divided peripheral CD4+ T cells following stimulation with anti-CD3 and anti-CD28 antibodies for 48 hr. We observed that the loss of LIS1 was associated with increased frequency of apoptotic cells among divided cells,

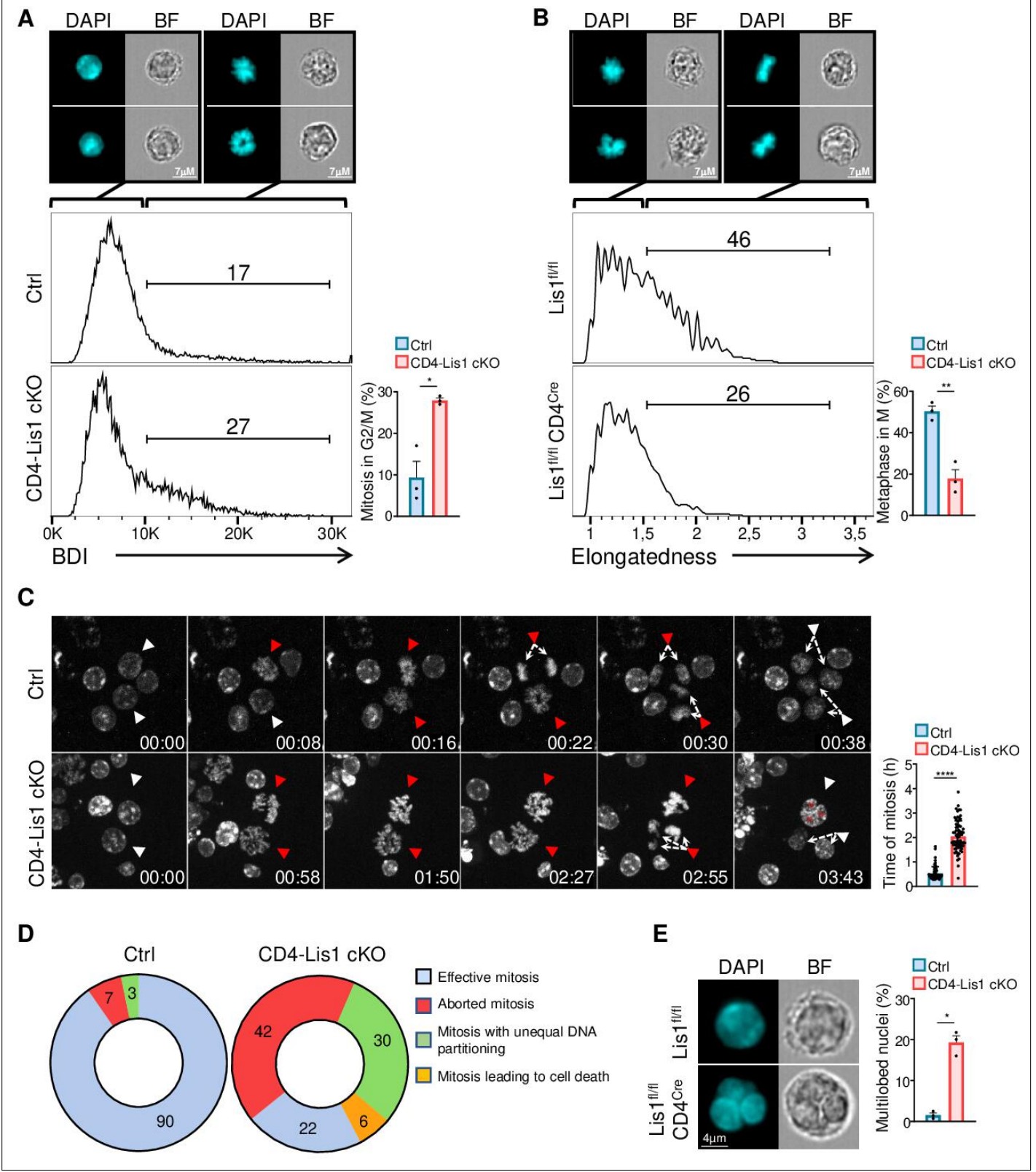

**Figure 4.** Dysfunctional chromosome alignment in LIS1-deficient CD4+ T cells leads to abortive mitosis and aneuploidy. (**A**) CD4+ T cells from control and CD4-Lis1 cKO mice were stimulated with anti-CD3 and anti-CD28 antibodies for 48 hr. Histogram graphs represent the *Bright Detail Intensity* (BDI) feature on CD4+ T cells in the G2/M phase as determined by image stream flow cytometry. Numbers represent the percentages of cells in mitosis according to the BDI feature. Images represent DAPI staining in BDI^low and BDI^hi control CD4+ T cells. Bar graphs represent the percentages of cells

*Figure 4 continued on next page*

*Figure 4 continued*

in mitosis (M) out of cells in the G2/M phase (n = 30,000 cells). Data are mean ± SD and represent three independent experiments each including n = 1 mouse per group. (**B**) CD4$^+$ T cells from control and CD4-Lis1 cKO mice were stimulated with anti-CD3 and anti-CD28 antibodies for 24 hr, synchronized with nocodazole for 18 hr and incubated with MG132 for 3 hr to induce metaphase arrest. Histogram graphs represent the *Elongatedness* feature on CD4$^+$ T cells in the M phase as determined by image stream flow cytometry. Numbers represent the percentages of cells in metaphase according to *Elongatedness* feature (n = 30,000 cells). Images represent DAPI staining in *Elongatedness*$^{low}$ and *Elongatedness*$^{hi}$ control CD4$^+$ T cells. Bar graphs represent the percentages of cells in metaphase out of cells in the M phase. Data are mean ± SD and represent three independent experiments each including n = 1 mouse per group. (**C**) Time-lapse microscopy analysis of cell division in CD4$^+$ T cells from control and CD4-Lis1 cKO mice stimulated with anti-CD3 and anti-CD28 antibodies. Images represent DNA staining on CD4+ T cells at the indicated times (hours:minutes). White arrows represent cells with uncondensed DNA. Red arrows represent the same cells after chromosomes formation. The top red arrows in the CD4-Lis1 cKO panel are representative of abortive mitosis. The bottom red arrows in the CD4-Lis1 cKO panel are representative of mitosis leading to aneuploidy. Bar graphs represent the time of mitosis per cell. Data are mean ± SD and represent three independent experiments each including n = 1 mouse per group. (**D**) Mitosis outcomes in control and CD4-Lis1 cKO CD4$^+$ T cells stimulated with anti-CD3 and anti-CD28 antibodies. Numbers represent percentages in the different section out of a total of n = 62–64 mitosis analyzed. Data represent three independent experiments each including n = 1 mouse per group. (**E**) CD4$^+$ T cells from control and CD4-Lis1 cKO mice were stimulated with anti-CD3 and anti-CD28 antibodies for 48 hr. Cells in G2 phase were analyzed by image stream flow cytometry. Cells stained with DAPI and bright-field (BF) images are represented. Bar graphs represent the percentages of cells with multilobed nuclei (n = 400 cells). Data are mean ± SD and represent three independent experiments each including n = 1 mouse per group. (**A, B**) Unpaired two-tailed Welch *t* tests were performed. (**C**) Unpaired two-tailed Mann–Whitney *t* test was performed. *p<0.05; **p<0.01; ***p<0.001; ****p<0.0001.

The online version of this article includes the following source data for figure 4:

**Source data 1.** Dysfunctional chromosome alignment in LIS1-deficient CD4+ T cells leads to abortive mitosis and aneuploidy.

---

but had no significant effect on apoptosis in activated CD25+ undivided cells (*Figure 5A*). Analysis of p53 expression prior the initial cycle of division at 24 hr showed comparable expression level of p53 between wild-type and LIS1-deficient cells, whereas p53 expression was dramatically increased in LIS1-deficient CD4+ T cells compared with that in control cells after the initial division cycles at 48 hr (*Figure 5B*). In comparison, the abundance of p53 after 48 hr of stimulation was comparable in wild-type and LIS1-deficient CD8+ T cells, supporting that the loss of LIS1 has a modest impact on cell division in the CD8+ lineage (*Figure 5B*). Analysis was next performed on DN3 thymocytes stimulated with OP9-Dl1 cells and led to a similar increase in apoptosis exclusively in divided thymocytes from LIS1-deficient mice (*Figure 5C*). The expression level of p53 was also strongly increased in total LIS1-deficient DN3 thymocytes compared with that in wild-type DN3 cells (*Figure 5D*). Altogether, those results indicate that the loss of LIS1 results in a defective chromosomes congression and separation during metaphase, which leads to aneuploidy, the upregulation of p53, and the development of apoptotic program.

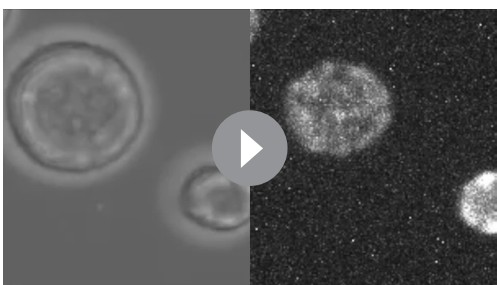

**Video 1.** Time-lapse microscopy of mitosis in wild-type CD4$^+$ T cells. Time-lapse microscopy analysis of mitosis in CD4$^+$ T cells from wild-type mice stimulated with anti-CD3 and anti-CD28 antibodies. Videos represent DNA staining (right panel) and bright field (left panel) on CD4+ T cells.

https://elifesciences.org/articles/80277/figures#video1

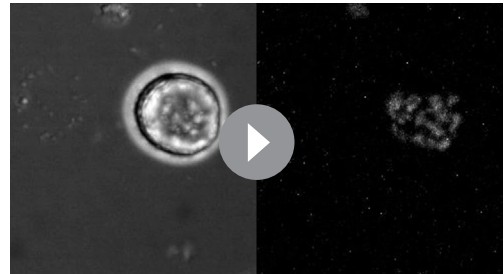

**Video 2.** Time-lapse microscopy of abortive mitosis in Lis1-deficient CD4$^+$ T cells. Time-lapse microscopy analysis of mitosis in CD4$^+$ T cells from CD4-Lis1 cKO mice stimulated with anti-CD3 and anti-CD28 antibodies. Videos represent DNA staining (right panel) and bright field (left panel) on CD4+ T cells.

https://elifesciences.org/articles/80277/figures#video2

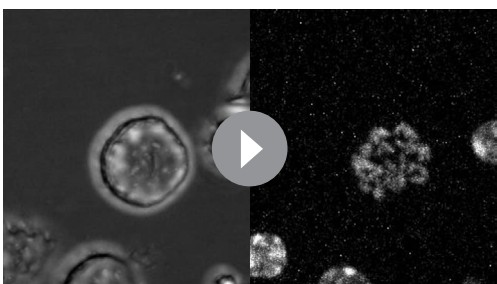

**Video 3.** Time-lapse microscopy of mitosis with aneuploidy in Lis1-deficient CD4+ T cells. Time-lapse microscopy analysis of mitosis in CD4+ T cells from CD4-Lis1 cKO mice stimulated with anti-CD3 and anti-CD28 antibodies. Videos represent DNA staining (right panel) and bright field (left panel) on CD4+ T cells.
https://elifesciences.org/articles/80277/figures#video3

# LIS1 controls mitotic spindle and centrosome integrity in CD4+ T cells by promoting the formation of dynein–dynactin complexes

Each spindle pole is normally established by one centrosome containing a pair of centrioles embedded in the pericentriolar material (PCM) containing γ-tubulin ring complexes (γ-TuRCs) from which microtubules nucleate. Centrosomes replicate once every cell cycle during the G1-S phase (*Nigg and Stearns, 2011*). Anomaly in centrosomes replication and PCM fragmentation may lead to the formation of extra-centrosomes that can be associated to the formation of multi-polar spindles and to the unequal repartition of chromosomes (*Holland and Cleveland, 2009*; *Maiato and Logarinho, 2014*). Previous studies in embryonic fibroblast show that the loss of LIS1 is associated with the formation of multipolar spindle due to the formation of extra-centrosomes (*Moon et al., 2014*). However, this defect is not systematically observed in the absence of LIS1. For instance, the loss of LIS1 in hematopoietic stem cells has a moderate effect on the integrity of the mitotic spindle but rather affects the spindle positioning during telophase, leading to increased rate of asymmetric divisions (*Zimdahl et al., 2014*).

To evaluate whether the loss of LIS1 could be associated with an aberrant number of centrosomes or a loss of centrosome integrity prior the division of CD4+ T cells, we stimulated CTV-stained CD4+ T cells from CD4-Lis1 cKO and control mice with anti-CD3 and anti-CD28 antibodies for 48 hr and FACS-sorted undivided CTV$^{hi}$ cells based on the forward-size-scattered parameter to discriminate unactivated (forward-scatter [FSC]$^{lo}$) from activated (FSC$^{hi}$) cells. Cells were analyzed by confocal microscopy after γ-tubulin and DAPI staining. In the presence of LIS1, we observed that the vast majority of FSC$^{lo}$ CD4+ T cells contained a single centrosome, whereas the majority FSC$^{hi}$ cells had two centrosomes as expected from cells in mitosis (*Figure 6A*). In the absence of LIS1, more than 50% of mitotic FSC$^{hi}$ CD4+ T cells had more than two centrosomes (*Figure 6A*). The loss of LIS1 did not affect centrosome copy numbers in unactivated CD4+ T cells (*Figure 6A*), indicating that LIS1 is engaged following TCR stimulation once the cell cycle has started, possibly at the stage of centrosome duplication. Some extra-centrosomes showed reduced accumulation of γ-tubulin compared with normal centrosomes in wild-type cells, suggesting that the loss of LIS1 leads to PCM fragmentation or to the loss of centrosome integrity rather than centrosome supernumerary duplication (*Figure 6B*). Analysis of γ- and α-tubulin stainings in LIS1-deficient CD4+ T cells show that these extra-centrosomes were 'active' in that they could effectively nucleate microtubule fibers (*Figure 6B*). Multiple centrosomes were also observed in cells-sorted post-β-selection DN3 thymocytes (*Figure 6C*). Together, these results indicate that LIS1 is required for the formation of stable bipolar mitotic spindles upon division of thymocytes and CD4+ T cells.

The biochemical basis by which LIS1 affects dynein function has been the focus of intense investigations yielding contradictory findings and divergent models (*Markus et al., 2020*). Evidence from early studies suggests that LIS1 might be acting as a 'clutch' that would prevent dynein's ATPase domain from transmitting a detachment signal to its track-binding domain (*Huang et al., 2012*). More recent in vitro investigations with recombinant proteins identify critical function for LIS1 in the assembly of active dynein–dynactin complexes (*Htet et al., 2020*; *Elshenawy et al., 2020*). To analyze whether the cellular defect observed in LIS1-deficient CD4+ T cells could be associated with defect in dynein–dynactin complex assembly, we compared the amount of p150Glued, a subunit of the dynactin complex, that co-immunoprecipitated with the intermediate chain of dynein (DIC) in CD4+ T cells isolated from CD4-Lis1 cKO and control mice (*Reck-Peterson et al., 2018*). The amount of p150Glued that co-immunoprecipitated with DIC was decreased in LIS1-deficient cells compared with wild-type controls (*Figure 6D*). Similar amount of the dynein heavy chain (DHC) was co-immunoprecipitated with the DIC in LIS1-deficient and wild-type cells (*Figure 6D*), indicating that the defect in

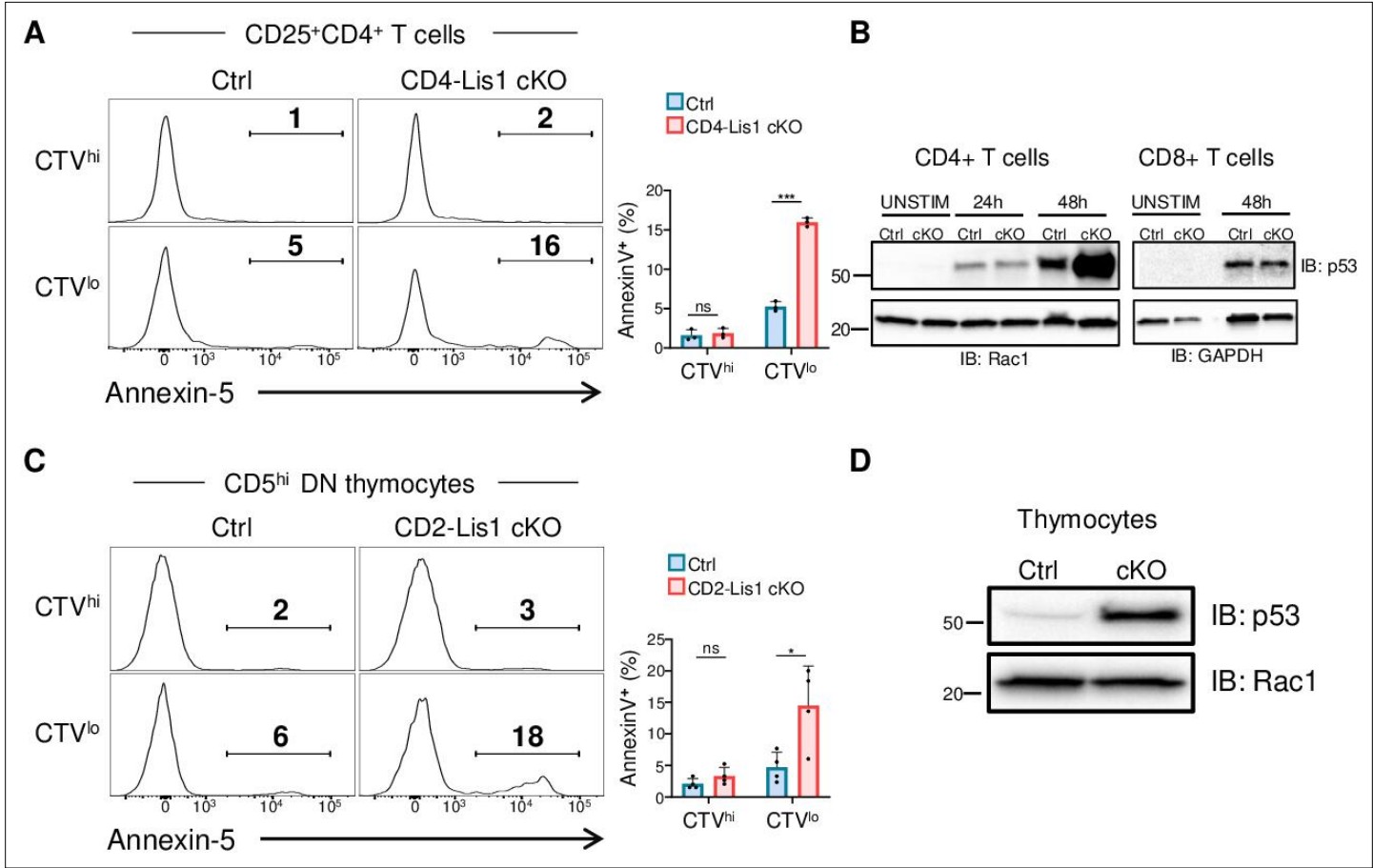

**Figure 5.** Proliferation leads to p53 upregulation and apoptosis in LIS1-deficient thymocytes and CD4+ T cells. (**A**) CD4+ T cells from control and CD4-Lis1 cKO mice were stained with CellTrace violet (CTV) and stimulated with anti-CD3 and anti-CD28 antibodies for 48 hr. The histogram graphs show annexin-5 staining on CTVhi (top panel) and CTVlow (bottom panel) CD25+CD4+T cells. Bar graphs represent the percentages of annexin5+ cells in the indicated subsets. Data are mean ± SD and represent two independent experiments each including n = 1–2 mice per group. (**B**) Total CD4+ and CD8+ T cells from control and CD4-Lis1 cKO mice were stimulated with anti-CD3 and anti-CD28 antibodies for the indicated times. Total cytoplasmic extracts of the cells were then analyzed by Western blotting with antibodies against p53, Rac1, and GAPDH, the loading controls. Data are representative of two independent experiments. (**C**) CD5lo DN3 thymocytes from control and CD2-Lis1 cKO mice were stained with CTV and stimulated with OP9-Dl1 cells for 48 hr. The histogram graphs show annexin-5 staining on CTVhi (top panel) and CTVlow (bottom panel) CD5hiCD4+ T cells. Bar graphs represent the percentages of annexin5+ cells in the indicated subsets. Data are mean ± SD and represent two independent experiments each including n = 2 mice per group. (**D**) Total cytoplasmic extracts of the DN thymocytes were analyzed by Western blotting with antibodies against p53 and Rac1, the loading control. Data are representative of two independent experiments. Unpaired two-tailed Welch t tests were performed in (**A, C**). *p<0.05; ***p<0.001.

The online version of this article includes the following source data for figure 5:

**Source data 1.** Proliferation leads to p53 upregulation and apoptosis in LIS1-deficient thymocytes and CD4+ T cells.

DIC-p150Glued interaction was not due do ineffective assembly of the dynein complex itself. These results suggest that LIS1 controls the integrity of mitotic spindle pole assembly in peripheral CD4+ T cells by stabilizing the association between dynein and dynactin complexes.

## Discussion

In this study, we identified a selective LIS1 requirement for mitosis in thymocytes and peripheral CD4+ T cells following β-selection and antigen priming, respectively. LIS1-dependent proliferation defects resulted in a block of early T-cell development and in a nearly complete lack of CD4+ T-cell expansion following activation. LIS1 deficiency in thymocytes and CD4+ T cells led to a disruption of dynein–dynactin complexes, which was associated with a loss of centrosome integrity and with the formation of multipolar spindles. These mitotic abnormalities were in turn associated to abnormal chromosomes

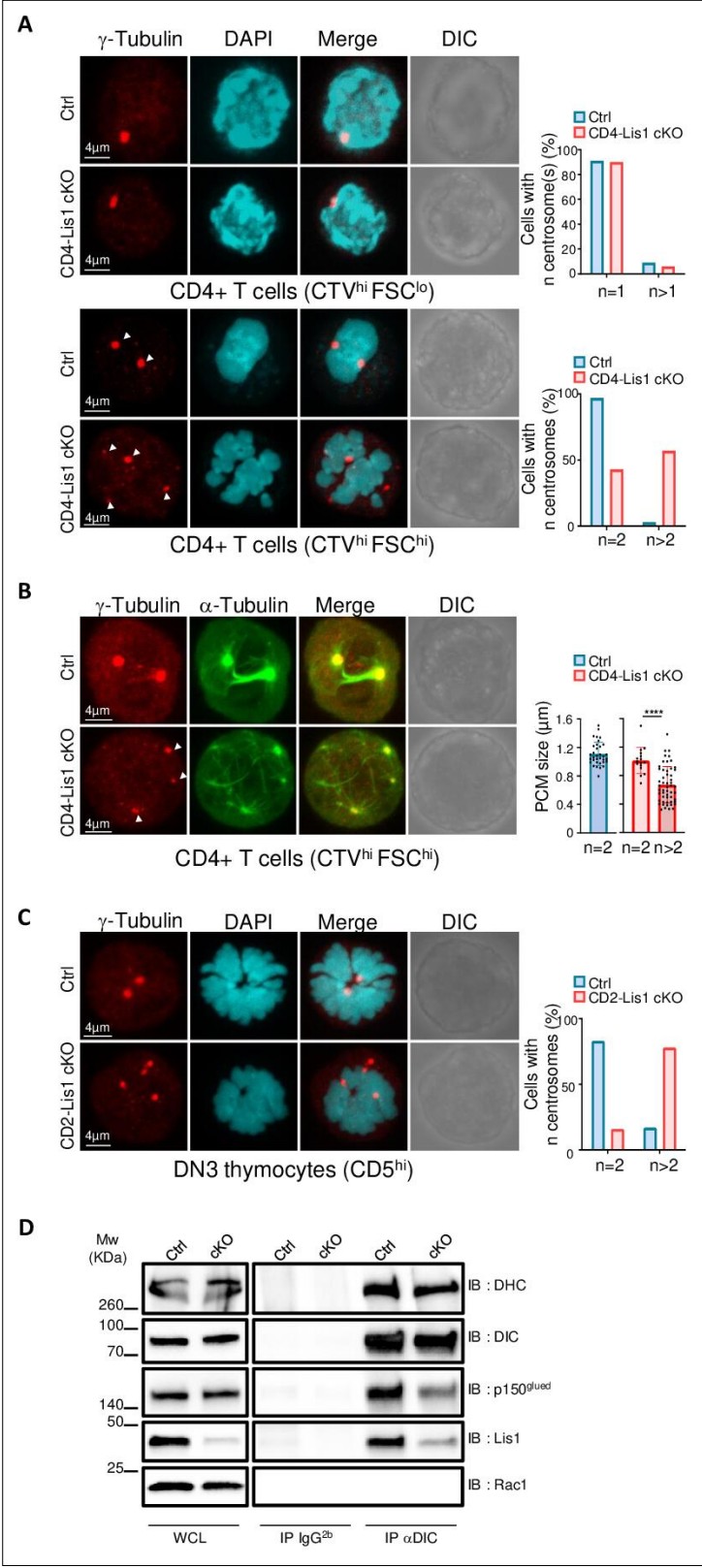

**Figure 6.** Impaired formation of dynein/dynactin complexes is associated with the loss of centrosome integrity and the formation of multipolar spindles in LIS-1-deficient thymocytes and CD4+ T cells. (**A**) CD4[+] T cells from control and CD4-Lis1 cKO mice were stained with CellTrace violet (CTV) and stimulated with anti-CD3 and anti-CD28 antibodies for 48 hr. Images represent maximum intensity projection of γ-tubulin and DAPI staining on undivided

*Figure 6 continued on next page*

*Figure 6 continued*

FSC$^{lo}$ (top panel) and FSC$^{hi}$ (bottom panel) CD4+ T cells. Bar graphs represent the percentages of cells with the indicated number of centrosome in total cells (top graph) or in mitotic cells (bottom graph). Data represent one experiment out of two independent experiments with n = 30–50 cells analyzed per group. (**B**) CD4$^{+}$ T cells from control and CD4-Lis1 cKO mice were stained with CTV and stimulated with anti-CD3 and anti-CD28 antibodies for 48 hr. Images represent maximum intensity projection of γ-tubulin and α-tubulin staining on undivided FSC$^{hi}$ CD4+ T cells. Bar graphs represent the size of the pericentriolar region (PCM) based on γ-tubulin staining in mitotic cells with the indicated number of centrosomes. Data represent three experiments with n = 16–54 centrosomes analyzed per group. (**C**) Images represent maximum intensity projection of γ-tubulin and DAPI staining CD5$^{hi}$ DN3 thymocytes. Bar graphs represent the percentages of cells with the indicated number of centrosomes in mitotic cells. Data represent one experiment out of two independent experiments with n = 30–50 cells analyzed per group. (**D**) CD4$^{+}$ T cell extracts from control and CD4-Lis1 cKO mice were subjected to immunoprecipitation (IP) with antibodies specific of the intermediate chain of dynein (DIC) or with an IgG2b isotype control and then analyzed by Western blotting with antibodies specific of the indicated proteins (dynein heavy chain [DHC]). Data represent one experiment out of two independent experiments. Unpaired two-tailed Mann–Whitney $t$ test was performed. ****p<0.0001.

The online version of this article includes the following source data for figure 6:

**Source data 1.** Impaired formation of dynein/dynactin complexes is associated with the loss of centrosome integrity and the formation of multipolar spindles in LIS-1 deficient thymocytes and CD4+ T cells.

reorganization during metaphase and telophase and to aneuploidy and p53 upregulation upon cell division. Importantly, whereas LIS1 deficiency led to a strong block of CD8+ T-cell proliferation upon PMA and ionomycin stimulation, it had a very little effect, if any, on the proliferation of CD8+ T cells following TCR engagement, suggesting that the mitotic machinery that orchestrates mitosis in CD8+ T cells upon TCR stimulation is different from that engaged in thymocytes and peripheral CD4+ T cells upon pre-TCR and TCR engagement.

LIS1 was shown to be dispensable for the proliferation of antigen-specific CD8+ T cell following infection with *L. monocytogenes* (*Ngoi et al., 2016*), supporting the data that we report here in CD8+ T cells following TCR stimulation. Comparable cell type-specific effects of LIS1 on proliferation have been described at early stages of neurogenesis and hematopoiesis (*Yingling et al., 2008*; *Zimdahl et al., 2014*). The loss of LIS1 in neuroepithelial stem cells leads to mitotic arrest and apoptosis upon symmetrical division events associated to progenitor cell maintenance, whereas it has only a moderate effect on asymmetrical division associated with neurogenesis, suggesting that symmetric division might be more LIS1-sensitive than asymmetric division (*Yingling et al., 2008*). Accordingly, LIS1 deficiency leads to a dramatic decrease in proliferation when CD8+ T cells are stimulated with soluble ligands such as cytokines and PMA/ionomycin, which favor symmetric division (*Ngoi et al., 2016*). This suggests that the different sensitivity of CD4+ and CD8+ T cells to LIS1 deficiency upon cell division is not simply the consequence of a preferential use of LIS1 in CD4+ T cells but rather the consequence of different mitotic organizations in CD4+ and CD8+ T cells in the context of polarized cell stimulations, which might exhibit different requirement for LIS1. This raises the question of whether CD4+ T cells would be more prone to symmetric divisions than CD8+ T cells. Theoretically, the experimental settings that we used in this study might not be optimal for eliciting asymmetric cell division since we stimulated T cells with anti-CD3 and anti-CD28 in the absence of ICAM-1, which is required for asymmetric cell divisions to occur in the context of APC stimulation (*Chang et al., 2007*). However, the rate of asymmetric cell divisions might be less influenced by ICAM-1 stimulation in conditions where plate-bound stimulations with antibodies are used (*Jung et al., 2014*). Asymmetric cell divisions have been detected in CD4+ T cells after the first antigen encounter (*Chang et al., 2007*), but it is unknown whether these divisions occur systematically or whether they occur with variable frequencies that could be context-dependent. It is also unclear whether CD4+ and CD8+ T cells have similar rates of asymmetric division since the literature lacks quantitative studies in which cellular events associated with mitosis would be investigated side-by-side in those two subsets. The repartition of the transcription factor T-bet in daughter cells was compared in one study by flow cytometry in CD4+ and CD8+ T cells after a first round of cell division (*Chang et al., 2011*). The authors showed that T-bet segregates unequally in daughter cells in both CD4+ and CD8+ T cells. However, the disparity of T-bet between daughter cells was higher in CD8+ T cells compared with that in CD4+ T cells (five- versus threefold), suggesting that cell-fate determinants are either more equally

(or less unequally) distributed in daughter cells from the CD4+ lineage or that the rate of symmetric divisions is higher in CD4+ T cells than in the CD8+ T cells. More extensive analysis would be required to precisely quantify the rate of symmetric and asymmetric cell divisions in CD4+ and CD8+ T cells in the context of APC stimulation.

Mechanistically, we show that LIS1 is important in CD4+ T cells to stabilize the interaction of the microtubule-associated motor protein dynein with the dynactin complex, which facilitates the binding of dynein to cargos and promotes thereby their transport along microtubule fibers. This is in agreement with recent in vitro studies showing that LIS1 is required for the efficient assembly of active dynein–dynactin complexes (*Htet et al., 2020*; *Elshenawy et al., 2020*). Given the pleiotropic role of the dynein–dynactin complexes during mitosis, several scenarios could possibly explain the defect of proliferation observed in thymocytes and peripheral CD4+ T cells. Two nonexclusive scenarios seem the most likely to us. A first scenario is that the loss of LIS1 leads to an inefficient attachment of the chromosome kinetochores to dynein, leading to metaphase delay and possibly asynchronous chromatid separation, a phenomenon called 'cohesion fatigue,' which leads to centriole separation and the formation of multipolar spindles (*Daum et al., 2011*). This possibility is supported by studies showing that LIS1 is localized to the kinetochores in fibroblasts and is required for the normal alignment of chromosomes during metaphases (*Moon et al., 2014*; *Faulkner et al., 2000*) and for targeting the dynein complex to kinetochore (*Moon et al., 2014*). A second possibility is that the absence of LIS1 leads to the fragmentation of the PCM, which is associated with the formation of multipolar spindles and the erroneous merotelic kinetochore-microtubule attachments (a single kinetochore attached to microtubules oriented to more than one spindle pole), which can cause chromosomal instability in cells that ultimately undergo bipolar division (*Cimini et al., 2001*). This is supported by the fact that several PCM components are transported toward centrosomes along microtubules by the dynein–dynactin motor complex (*Bärenz et al., 2011*; *Dammermann and Merdes, 2002*) and that the depletion of multiple pericentriolar proteins results in PCM fragmentation, which subsequently generates multipolar spindles (*Dammermann and Merdes, 2002*; *Krauss et al., 2008*; *Kim and Rhee, 2011*).

We previously identified LIS1 as a binding partner of the signaling protein THEMIS in thymocytes and confirmed this interaction through yeast two-hybrid approaches (*Zvezdova et al., 2016*; *Garreau et al., 2017*). THEMIS enhances positive selection in thymocytes (*Fu et al., 2009*; *Lesourne et al., 2009*; *Johnson et al., 2009*) and is important for the maintenance of peripheral CD8+ T cells by stimulating cytokine-driven signals leading to homeostatic proliferation (*Brzostek et al., 2020*). Although LIS1 deficiency does not modulate the efficiency of thymocyte-positive selection, the loss of LIS1 is associated with a strong defect of peripheral T-cell proliferation in response to IL-2 and IL-15 stimulation (*Ngoi et al., 2016*). THEMIS and LIS1 deficiencies both lead to severely compromised CD8+ T-cell proliferation following transfer in lymphopenic hosts (*Ngoi et al., 2016*; *Brzostek et al., 2020*). Although this defect was attributed to stimulatory function of THEMIS on IL-2- and IL-15-mediated signaling, we cannot rule out the possibility that THEMIS would play a more direct role in cell cycle by controlling LIS1-mediated events. THEMIS operates by repressing the tyrosine phosphatase activity of SHP-1 and SHP-2, which are key regulatory proteins of TCR signaling (*Choi et al., 2017*). Gain-of-function mutations of SHP-2 in mouse embryonic fibroblast and leukemia cells lead to centrosome amplification and aberrant mitosis with misaligned chromosomes (*Liu et al., 2016*). Thus, the hyper activation of SHP-2 resulting from THEMIS deficiency may lead to cellular defects similar to those observed in LIS1-deficient T cells. An interesting perspective to this work would be to investigate further whether the loss of THEMIS in CD8+ T cells would lead to similar mitotic defects to those observed in LIS1-deficient thymocytes and CD4+ T cells upon TCR stimulation.

The fact that LIS1 deficiency increases the frequency of aneuploidy and leads to the upregulation of p53 expression suggests that defects affecting LIS1 expression or function could favor oncogenic transformation in lymphoid cells. LIS1 is necessary for the extensive growth of tumor cells in some cancer models. The genetic disruption of LIS1 in hematopoietic stem cells blocks the propagation of myeloid leukemia (*Zimdahl et al., 2014*). However, several evidence suggests also that the alteration of LIS1 expression could contribute to the carcinogenesis of several cancers such as hepatocellular carcinoma (*Li et al., 2018*; *Xing et al., 2011*), neuroblastoma (*Messi et al., 2008*), glioma (*Suzuki et al., 2007*), and cholangiocarcinoma (*Yang et al., 2014*). Thus, although a minimal expression level of LIS1 might be mandatory for extensive tumor growth, partial deficiencies in LIS1 might favor oncogenic transformation. Although monoallelic deficiency of LIS1 did not detectably affect CD4+ T-cell

proliferation in vitro, the partial loss of LIS1 function may enhance the risk of aneuploidy-driven cancer in a tumor-suppressor-failing context. This could be relevant in humans since genetic variants on *Pafah1b1* have been associated with a higher risk of developing acute myeloid leukemia (*Cao et al., 2017*).

# Materials and methods

**Key resources table**

| Reagent type (species) or resource | Designation | Source or reference | Identifiers | Additional information |
|---|---|---|---|---|
| Genetic reagent (*Mus musculus*) | 129S-Pafah1b1<sup>tm2Awb</sup>/J | Jackson Laboratories | Strain #:008002; RRID:IMSR_ JAX:008002 | This stain was provided by Dr. Deanna S. Smith (University of South Carolina, Columbia, USA) |
| Genetic reagent (*Mus. musculus*) | B6.Cg-Tg(CD2-icre)4Kio/J | Jackson Laboratories | Strain #:008520; RRID:IMSR_ JAX:008520 | |
| Genetic reagent (*M. musculus*) | Tg(Cd4-cre)1Cwi/BfluJ | Jackson Laboratories | Strain #:017336; RRID:IMSR_ JAX:017336 | |
| Cell line (*M. musculus*) | OP9-dl1 | *Schmitt et al., 2004* | | Provided by Dr. Sophie Laffont Pradines (Toulouse Institute for Infectious and Inflammatory Diseases, Toulouse France) |
| Antibody | Anti-CD3ε (hamster monoclonal) | BioLegend | Clone 2C-11 | Purified unconjugated |
| Antibody | Anti-CD28 (hamster monoclonal) | BioLegend | Clone 37.51 | Purified unconjugated |
| Antibody | Anti-CD8α (rat monoclonal) | Thermo Fisher Scientific | Clone 53-6.7 | Conjugated to A-700 (1/300) |
| Antibody | Anti-CD4 (rat monoclonal) | BD Biosciences | Clone RM4-5 | Conjugated to Pacific Blue (1/1000) |
| Antibody | Anti-CD24 (rat monoclonal) | BioLegend | Clone M1/69 | Conjugated to PE (1/500) |
| Antibody | Anti-TCRβ (hamster monoclonal) | BD Biosciences | Clone H57-597 | Conjugated to FITC (1/400) |
| Antibody | Anti-TCRβ (hamster monoclonal) | Thermo Fisher Scientific | Clone H57-597 | Conjugated to PECy7 (1/1500) |
| Antibody | Anti-Vα11 (rat monoclonal) | BD Biosciences | Clone RR8-1 | Conjugated to FITC (1/400) |
| AntibodyA | Anti-CD5 (rat monoclonal) | BD Biosciences | Clone 53-7.3 | Conjugated to APC (1/1000) |
| Antibody | Anti-CD5 (rat monoclonal) | Thermo Fisher Scientific | Clone 53-7.3 | Conjugated to FITC (1/1000) |
| Antibody | Anti-CD69 (hamster monoclonal) | BD Biosciences | Clone H1.2F3 | Conjugated to FITC (1/200) |
| Antibody | Anti-B220 (rat monoclonal) | BD Biosciences | Clone RA3-6B2 | Conjugated to PE (1/400) |
| Antibody | Anti-Gr1 (rat monoclonal) | BioLegend | Clone RB6-8C5 | Conjugated to PE (1/300) |

*Continued on next page*

*Continued*

| Reagent type (species) or resource | Designation | Source or reference | Identifiers | Additional information |
|---|---|---|---|---|
| Antibody | Anti-CD11b (rat monoclonal) | BioLegend | Clone M1/70 | Conjugated to PE (1/200) |
| Antibody | Anti-CD11c (hamster monoclonal) | BioLegend | Clone N418 | Conjugated to PE (1/200) |
| Antibody | Anti-Ter119 (rat monoclonal) | BioLegend | Clone TER119 | Conjugated to PE (1/200) |
| Antibody | Anti-CD3ε (hamster monoclonal) | BioLegend | Clone 145-2C11 | Conjugated to PE (1/200) |
| Antibody | Anti-NK1.1 (mouse monoclonal) | BD Biosciences | Clone PK136 | Conjugated to PE (1/200) |
| Antibody | Anti-TCRγδ (hamster monoclonal) | BD Biosciences | Clone GL3 | Conjugated to (1/200) |
| Antibody | Anti-CD44 (rat monoclonal) | Thermo Fisher Scientific | Clone IM7 | Conjugated to FITC (1/200) |
| Antibody | Anti-CD25 (rat monoclonal) | BD Biosciences | Clone PC61.5 | Conjugated to PercP Cy5.5 (1/300) |
| Antibody | Anti-CD71 (rat monoclonal) | BioLegend | Clone R17217 | Conjugated to PeCy7 (1/400) |
| Antibody | Anti-CD27 (hamster monoclonal) | BD Biosciences | Clone LG.3A10 | Conjugated to APC (1/200) |
| Antibody | Anti-IL-7R (rat monoclonal) | BD Biosciences | Clone A7R34 | Conjugated to A700 (1/500) |
| Antibody | Anti-IL-7R (rat monoclonal) | BD Biosciences | Clone A7R34 | Conjugated to APC (1/400) |
| Antibody | Anti-BCL-2 (hamster monoclonal) | BD Biosciences | Clone 3F11 | Conjugated to FITC (5 µL/$10^5$ cells) |
| Antibody | Anti-CD19 (rat monoclonal) | BioLegend | Clone 1D3/CD19 | Conjugated to PercPCY5.5 (1/500) |
| Antibody | Anti-c-kit (rat monoclonal) | BioLegend | Clone 2B8 | Conjugated to PE (1/200) |
| Antibody | Anti-c-kit (rat monoclonal) | BD Biosciences | Clone 2B8 | Conjugated to APC (1/200) |
| Antibody | Anti-IgM (rat monoclonal) | BD Biosciences | Clone RMM-1 | Conjugated to PECy7 (1/300) |
| Antibody | Anti-CD45.1 (mouse monoclonal) | BD Biosciences | Clone A20 | Conjugated to PE (1/500) |

*Continued on next page*

*Continued*

| Reagent type (species) or resource | Designation | Source or reference | Identifiers | Additional information |
|---|---|---|---|---|
| Antibody | Anti-γ-tubulin (mouse monoclonal) | BioLegend | Clone 14C11 | Purified unconjugated |
| Antibody | Anti-α-tubulin (mouse monoclonal) | Thermo Fisher Scientific | Clone DM1A | Purified unconjugated |
| Antibody | Goat anti-mouse IgG2b | Thermo Fisher Scientific | Cat#A-21147 | Alexa Fluor 555 |
| Antibody | Anti-Dynein IC (mouse monoclonal) | Santa Cruz Biotechnologies | Clone 74-1 | Purified unconjugated |
| Antibody | Anti-LIS1 (rabbit polyclonal) | Santa Cruz Biotechnologies | sc-15319 | Purified unconjugated |
| Antibody | Anti-Dynein HC (rabbit polyclonal) | Santa Cruz Biotechnologies | sc-9115 | Purified unconjugated |
| Antibody | Anti-p150glued (mouse monoclonal) | BD Biosciences | Clone 1/p150Glued | Purified unconjugated |
| Antibody | Anti-p53 (mouse monoclonal) | Cell Signaling | Clone 1C12 | Purified unconjugated |
| Antibody | Anti-Rac1 (mouse monoclonal) | Millipore | Clone 23A8 | Purified unconjugated |
| Other | AnnexinV | BD Biosciences | RRID:AB_2868885 | APC ($5\ \mu L/10^5$ cells) |
| Other | AnnexinV binding buffer | BD Biosciences | Cat#556454 | Used for annexinV staining |
| Other | eBioscience Fixable Viability Dye | Thermo Fisher Scientific | Cat#65-0865-14 | eFluor 780 APC-H7 |
| Other | Permeabilization buffer | Thermo Fisher Scientific | Cat#00-8333-56 | Used for intracytoplasmic staining |
| Other | Chambered glass coverslip | IBIDI | Cat#80821 | Used for videomicroscopy analyses |
| Other | Dynabeads Untouched Mouse CD4 Cells Kit | Thermo Fisher Scientific | Cat#11415D | Magnetic beads used for the purification of CD4-CD8- thymocytes as well as CD4+ and CD8+ T cells |
| Other | DAPI | Sigma-Aldrich | Cat#D9542 | 1 mg/mL Nuclear staining for microscopy |
| Other | Hoechst 33342 | Sigma-Aldrich | Cat#14533 | 50 ng/mL Nuclear staining for videomicroscopy |
| Other | Cell trace Violet | Thermo Fisher Scientific | Cat#C34557 | 2 μM Cell tracker used for proliferation analyses |
| Other | DABCO | Sigma-Aldrich | Cat#D27802 | Mounting medium for microscopy |
| Other | Mouse IL-7 | PeproTech | Cat#21--17 | 10 ng/mL |
| Chemical compound, drug | Nocodazole | Sigma-Aldrich | Cat#M1404 | 100 ng/mL Inhibitor of microtubule polymerization |
| Chemical compound, drug | MG132 | Sigma-Aldrich | Cat#M7449 | 10 μM proteasome inhibitor |
| Chemical compound, drug | Phorbol 12-myristate 13-acetate (PMA) | Sigma-Aldrich | Cat#P8139 | 100 ng/ml T-cell pharmacological stimulator |
| Chemical compound, drug | Ionomycin | Sigma-Aldrich | Cat#I0634 | 100 ng/ml T-cell pharmacological stimulator |
| Chemical compound, drug | RIBI | Sigma Adjuvant System | Cat#S6322 | Adjuvant |
| Software, algorithm | IDEAS | Millipore | | |

## Mice

*Pafah1b1*<sup>flox/flox</sup> mice were described previously (*Hirotsune et al., 1998*). These mice were bred with *Cd2-cre* transgenic mice (https://www.jax.org/strain/008520) in which the human *cd2* promoter directs the expression of the CRE recombinase at early stages of T- and B-cell development. *Pafah1b-1*<sup>flox/flox</sup> mice were also bred with *Cd4-Cre* transgenic mice (https://www.jax.org/strain/017336) in which the *cd4* promoter directs the expression of the CRE recombinase during T-cell development in CD4+CD8+ thymocytes. AND and OT-2 TCR-transgenic mice were from Taconic Farms. All the experiments were conducted with sex and age-matched mice between 6 and 12 weeks old housed under specific pathogen-free conditions at the INSERM animal facility (US-006; accreditation number A-31 55508 delivered by the French Ministry of Agriculture to perform experiments on live mice). All experimental protocols were approved by a ministry-approved ethics committee (CEEA-122) and follow the French and European regulations on care and protection of the Laboratory Animals (EC Directive 2010/63).

## Antibodies

The following antibodies were used.

*For stimulation and cell culture*: anti-CD3ε (145-2C11) and anti-CD28 (37.51) antibodies were from BioLegend. *For cell sorting and flow cytometry analysis:* anti-CD8α (clone 53-6.7), anti-CD4 (clone RM4-5), anti-CD24 (clone M1/69), anti-TCRβ (clone H57-597), anti-Vα11 (clone RR8-1), anti-CD5 (clone 53-7.3), anti-CD69 (clone H1.2F3), anti-B220 (clone RA3-6B2), anti-Gr1 (clone RB6-8C5), anti-CD11b (clone M1/70), anti-CD27 (clone LG.3A10), anti-CD11c (clone N418), anti-Ter119 (clone TER119), anti-CD3 (clone 145-2C11), anti-NK1.1 (clone PK136), anti-TCRγδ (clone GL3), anti-CD44 (clone IM7), anti-CD25 (clone PC61.5), anti-CD71 (clone R17217), anti-IL-7R (clone A7R34), anti-BCL-2 (clone 3F11), anti-CD19 (clone 1D3/CD19), anti-c-kit (clone 2B8), anti-IgM (clone RMM-1), and anti-CD45.1 (clone A20) were from BD Biosciences and BioLegend. *For imaging studies:* anti-γ-tubulin (clone 14C11) was from BioLegend and anti-α-tubulin (DM1A) was from (Thermo Fisher Scientific). *For immunoprecipitation and Western blot analysis:* anti-DIC (clone 74-1), IgG2b isotype control (sc-3879), anti-LIS1 (sc-15319), and anti-DHC (sc-9115) were from Santa Cruz Biotechnologies. Anti-p150glued (clone 1/p150Glued) were from BD Biosciences, anti-p53 (clone 1C12) were from Cell Signaling, and anti-Rac1 (clone 23A8) were from Millipore.

## Flow cytometry and cell sorting

For flow cytometry analysis, single-cell suspensions from thymus, spleen, lymph nodes, and bone marrows were incubated with diluted eBioscience Fixable Viability Dye eFluor 780 (Thermo Fisher) in phosphate-buffered saline (PBS) prior staining with fluorochrome-conjugated antibodies. Intracellular staining was performed after cell fixation with 4% paraformaldehyde (PFA) by incubating the cells with conjugated antibodies in permeabilization buffer (Thermo Fisher Scientific). For the phenotyping of DN subsets, thymocytes were stained with an anti-lineage cocktail (anti-Gr1, anti-CD11b, anti-CD11c, anti-Ter119, anti-CD3, anti-B220, anti-NK1.1, and anti-TCRγδ) and with anti-CD8α and anti-CD4 antibodies. Data acquisition was performed on a BD LSRII flow cytometer and analysis with the FlowJo software.

For DN3 cell purification, thymocytes were first immunomagnetically depleted of CD3-, CD4-, or CD8α-positive cells. Lin⁻CD44⁻CD25⁺CD5⁻ or Lin⁻CD44⁻CD25⁺CD71⁻ DN3 cells were sorted on a BD FACS Aria cell sorter. For peripheral T-cell isolation, total CD4+ T cells and CD8+ T cells were purified from ACK-treated pooled lymph nodes and spleen by magnetic immunodepletion of CD8⁺, B220⁺, MHCII⁺, NK1.1⁺, Fcγ⁺, and CD11b⁺ cells and CD4⁺, B220⁺, MHCII⁺, NK1.1⁺, Fcγ⁺, and CD11b⁺ cells, respectively.

## Cell culture

The OP-9-dl1 cell line was provided by Dr. Sophie Laffont-Pradines (Toulouse Institute for Infectious and Inflammatory Diseases, Toulouse, France) and was initially generated in the group of Dr. Juan-Carlos Zúñiga-Pflücker (Sunnybrook Research Institute, Toronto, Ontario, Canada). This fibroblast cell line has been originally transfected by Notch ligand and can be authenticated by its exclusive capacity to stimulate progenitor T-cell differentiation in vitro (*Schmitt et al., 2004*). The cell line was tested negative for mycoplasma. OP9-DL1 cells were seeded at 8000 cells per well in 48-well plates

and incubated for 24 hr in OP9 culture media (alpha-MEM, 20% FCS, Penicillin and Streptomycin), followed by addition of 100,000 sorted CD5⁻ or CD71⁻ DN3 thymocytes per well together with 10 ng/mL recombinant mouse IL-7 (PeproTech).

For proliferation analysis, CD5⁻ DN3 thymocytes, CD4+ and CD8+ lymph nodes T cells were labeled with 2 µM CTV (Thermo Fisher Scientific) for 15 min at 37°C. Thymocytes were cultured with OP9-DL1 cells and peripheral T cells were incubated with the indicated doses of anti-CD3 antibodies and with 2 µg/mL anti-CD28 antibodies for 48 and 72 hr. For apoptosis analysis, thymocytes and CD4+ T cells were stained with CTV and stimulated for 48 hr as described for proliferation analysis. After stimulation, cells were stained with fluorochrome-conjugated annexin-5 (BD Biosciences) in annexin-5 binding buffer (BD Biosciences). For cell-cycle analysis, thymocytes and CD4+ T cells were stimulated for 48 hr as indicated above. Cells were fixed with 4% PFA and incubated with permeabilization buffer prior staining with DAPI in PBS.

## Image stream flow cytometry

For the analysis of the G2/M population, CD4+ T cells were stimulated with 10 µg/mL of anti-CD3 antibodies with 2 µg/mL of anti-CD28 antibodies for 24 hr. Cells were synchronized by addition of nocodazole (Sigma-Aldrich) at 100 ng/mL for 18 hr. Cells were then washed in RPMI supplemented with 10% FCS and incubated with 10 µM of MG132 (Sigma-Aldrich) for 3 hr. Cells were incubated with Fixable Viability Dye prior staining with fluorochrome-conjugated anti-CD4 antibodies and DAPI and acquired on an ImageStreamX apparatus from Millipore.

Data were analyzed using the IDEAS analysis software from Millipore. We used the 'Bright Detail Intensity' (BDI) parameter to discriminate mitotic cells from cells in the G2 phase. This parameter calculates the intensity of the bright pixels after subtraction of the background noise from the images. Cells in mitosis having condensed DNA will present a homogeneously bright staining leading to higher BDI value than cells in the G2 phase with uncondensed DNA. To evaluate cells in metaphase, we used the parameter 'Elongatedness,' which calculates the length to width ratio on a predefined DAPI mask. Cells with an 'Elongatedness' value exceeding 1.5 were characterized as cells in metaphase.

## Immunization with ovalbumin

CD45.1+CD4+ T cells were purified from lymph nodes and splenocytes from control and CD4-Lis1 cKO mice expressing the OT2 TCR. $2 \times 10^6$ cells in PBS were injected i.v. into C57BL/6J mice (CD45.2⁺) 1 hr before immunization with 40 µg of ovalbumin emulsified with RIBI (Sigma Adjuvant System). CD4⁺ T cell populations from the spleen were analyzed 2 and 3 days after immunization.

## Confocal analysis

CD4+ T cells were labeled with CTV and incubated with 10 µg/mL of anti-CD3 antibodies and 2 µg/mL of anti-CD28 antibodies for 48 hr. The CTVʰⁱFSCˡᵒ (non-proliferating, non-activated) and CTVʰⁱFSCʰⁱ (non-proliferating, activated) populations were sorted by flow cytometry. Lin⁻CD44⁻CD25⁺CD5ʰⁱ thymocytes were sorted by flow cytometry. Cells were deposited on 0.01% poly-L-lysine adsorbed slides (Sigma-Aldrich), fixed with 4% PFA, and permeabilized in PBS containing 0.1% Saponin (Sigma-Aldrich). α- and γ-Tubulin staining was made in PBS containing 0.1% saponin, 3% bovine serum albumin (BSA), and 10 mM HEPES at 4°C for 18 hr and revealed with fluorochrome-conjugated anti-mouse and IgG1 and IgG2b antibodies (Thermo Fisher Scientific) for 1 hr at room temperature. DNA was stained with DAPI for 15 min at room temperature in PBS. The slides were then mounted with DABCO solution (Sigma-Aldrich), and the images were acquired with an LSM710 confocal microscope equipped with a 63× 1.4 NA objective (Zeiss).

For video microscopy, CD4+ T cells were cultured with 10 µg/mL of anti-CD3 and 2 µg/mL of anti-CD28 antibodies on a chambered glass coverslip (IBIDI) for 24 hr. To stain DNA, Hoechst 33342 (Sigma-Aldrich) was added to the culture at a final concentration of 50 ng/mL. Cells were observed for 18 hr in a chamber at 37°C and 5% $CO_2$ with a Spinning disk confocal microscope. The z-stack images were edited into film and analyzed using ImageJ.

## Immunoprecipitation and Western blot analysis

For immunoprecipitation, CD4+ T cells were resuspended in 2 mL of ice-cold lysis buffer (10 mM Tris-HCl pH 7.4, 150 mM NaCl, 1% Triton, 2 mM Na₃VO₄, 5 mM NaF, 1 mM EDTA, and protease

inhibitor cocktail tablet [Roche]) and incubated for 20 min on ice. Lysates were cleared by centrifugation at 18,000 × *g* for 15 min at 4°C, and the dynein intermediate chain (DIC) was subjected to immunoprecipitation from cleared lysates for 2 hr at 4°C with 15 µL of protein G-Sepharose resin coated with 12 µg of polyclonal rabbit anti-DIC antibodies. The resin was washed three times and incubated for 10 min at 95°C with Laemmli buffer. For p53 analysis, CD4+ T cells were stimulated with 10 µg/mL of anti-CD3 and 2 µg/mL of anti-CD28 antibodies for 24 and 48 hr and were suspended in ice-cold lysis buffer after each time point. Proteins were resolved by SDS-PAGE and transferred to PVDF membranes according to standard protocols. Membranes were blocked with 5% milk in Tris-buffered saline containing Tween at 0.05% for 1 hr at room temperature before being incubated with primary antibodies at 4°C overnight. After washing, membranes were incubated with secondary antibodies for 1 hr at room temperature. Subsequently, membranes were incubated with enhanced chemiluminescence solution (Sigma) for 5 min in the dark, and luminescence was captured with a Bio-Rad XRS+ imager.

## Statistical analysis

GraphPad Prism was used to perform statistical analysis. All values in the article are presented as mean ±SD. Except when indicated, statistical significance was calculated by unpaired two-tailed Mann–Whitney *t* test. *p<0.05, **p<0.001, ***p<0.0001, ****p<0.0001.

## Acknowledgements

The authors thank Loïc Dupré for helpful comments and suggestions and Dr. Deanna Smith for providing *Pafah1b1*flox/flox mice. We acknowledge the technical assistance provided by the personnel of INSERM US006 Anexplo/creffre animal facility. The authors thank Fatima-Ezzahra L'Faqihi-Olive and Anne-Laure Iscache from the cytometry facility of INFINITy as well as Sophie Allart and Astrid Canivet from the cell imaging facility of INFINITy. This work was supported by INSERM; the Foundation ARSEP; the Association pour la Recherche sur le Cancer (ARC); the Agence Nationale de la Recherche (ANR-20-CE15-0002); the French Ministry of Higher Education and Research (PhD fellowship for JA and SM).

## Additional information

### Funding

| Funder | Grant reference number | Author |
|---|---|---|
| The French Ministry of Higher Education and Reserach | PhD fellowship | Jérémy Argenty<br>Suzanne Mélique |
| Association pour la recherche sur la Sclérose en Plaques | | Renaud Lesourne |
| Agence Nationale de la Recherche | ANR-20-CE15-0002 | Renaud Lesourne |
| Association pour la Recherche sur le Cancer | | Renaud Lesourne |

The funders had no role in study design, data collection and interpretation, or the decision to submit the work for publication.

### Author contributions

Jérémy Argenty, Conceptualization, Data curation, Formal analysis, Investigation; Nelly Rouquié, Data curation, Formal analysis, Investigation; Cyrielle Bories, Formal analysis, Investigation; Suzanne Mélique, Data curation, Formal analysis; Valérie Duplan-Eche, Software, Formal analysis, Methodology; Abdelhadi Saoudi, Supervision, Writing - review and editing; Nicolas Fazilleau, Supervision, Methodology, Writing - review and editing; Renaud Lesourne, Conceptualization, Supervision, Funding acquisition, Writing - original draft, Writing - review and editing

## Author ORCIDs

Abdelhadi Saoudi http://orcid.org/0000-0001-7015-8178
Renaud Lesourne http://orcid.org/0000-0003-3816-7087

## Ethics

All the experiments were conducted at the INSERM animal facility (US-006; accreditation number A-31 55508 delivered by the French Ministry of Agriculture to perform experiments on live mice). All experimental protocols were approved by a Ministry-approved ethics committee (CEEA-122) and follow the French and European regulations on care and protection of the Laboratory Animals (EC Directive 2010/63).

## Decision letter and Author response

Decision letter https://doi.org/10.7554/eLife.80277.sa1
Author response https://doi.org/10.7554/eLife.80277.sa2

## Additional files

### Supplementary files

• MDAR checklist

### Data availability

All data generated or analysed during this study are included in the manuscript and supporting file have been provided for Figures 1 and 3.

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
