## [Editor Report]

This paper reports a fundamental finding that the requirement for the dynein binding protein, Lis1, is differentially required by different T cell lineages. The paper provides compelling evidence that the regulation of cell division and subsequent fate differs across T cell lineages. This work will be of interest to researchers focussed on T cell biology and mitosis.

---

## [Decision Letter]

**Decision letter after peer review:**

Thank you for submitting your article "A selective LIS1 requirement for mitotic spindle assembly discriminates distinct T-cell division mechanisms within the T-cell lineage" for consideration by *eLife*. Your article has been reviewed by 3 peer reviewers, and the evaluation has been overseen by a Reviewing Editor and Tadatsugu Taniguchi as the Senior Editor. The reviewers have opted to remain anonymous.

Essential revisions:

Critical controls:

1. Control for the impact of Cre expression (and perhaps LIS1flox+/-).

2. Assess the ablation of Lis1 to ensure that the effects (and particularly the difference in effect between CD4 and CD8) are not an artefact of the experimental approach to deletion.

Critical extensions of the work:

3. More experimentation to clarify the step in β-selection that are disrupted by LIS deletion (expression of Notch and IL7R signalling components, cell cycle analysis, assess DN3a and DN3b stages).

4. Extend analysis of CD4 responses to later timepoints.

Modifications of interpretation:

5. Incorporate findings In the literature into the discussion on ACD as an explanation of the differences in proliferative effects between CD4 and CD8 cells.

*Reviewer #1 (Recommendations for the authors):*

a) It has been shown in the literature (e.g. https://doi.org/10.4049/jimmunol.1600827) that Cre itself can affect T cell differentiation. In this publication, the authors compared Cd2-Cre Lisflox/flox versus Lisflox/flox or Cd4-Cre Lisflox/flox versus Lisflox/flox, which means it is difficult to be sure that the effect is entirely due to the deletion of Lis1 and not partly due to the Cre itself. I think that it is important that they include this control.

b) In figure 1 it would be helpful to determine the effect of Lis1 deletion on icβTCR DN3a and DN3b cells, as opposed to DN3 cells, as the low levels of icβTCR at the DN3a stage may be obscuring the effect of Lis1.

c) In Cd4-Cre Lisflox/flox mice there was a drop in peripheral CD4^+^ and CD8^+^ cells and a 10% drop in proliferating CD8^+^ cells. The authors claim that this demonstrates that Lis1 is therefore not required for the proliferation of CD8 cells. However, as there is a slight effect, I do not feel that you can definitively claim that there is no effect. It would be useful to perform the cell cycle analysis and determine whether there is a defect in cell division, or not, to support this claim. Also, I am unclear as to why there was a drop in peripheral CD8^+^ cells if Lis1 did not affect proliferation.

d) In figure 4C, D you state that some cells are divided with an unequal repartition of the chromosomes in the daughter cells. Please state how this was measured.

e) In some of the figure legends the authors state: Data represent one experiment out of two independent experiments with n=30-50 cells analyzed per group. Please clarify why you haven't shown the data from both experiments.

f) In the discussion there is a paragraph correlating the lack of effect on CD8 cells and asymmetric cell division. I think this is an interesting point and while I appreciate that is it possibly beyond the scope of this paper I do think that the authors should draw on the literature examining the occurrence of asymmetric cell division in CD4 cells versus CD8 cells to strengthen this section of the discussion.

*Reviewer #2 (Recommendations for the authors):*

Lis1 is a dynein binding protein previously shown to be important for some types of T cell proliferation including homeostatic proliferation, but less important for the response of CD8 T cells to TCR stimulation. However, it is required for the development of CD8 T cell memory in model systems. In the present study, the authors repeat and extend the previous work. They construct iCD2-Cre mice harboring a conditional allele of Lis1 that it is critically required for proliferation in T cell development. Repeating earlier experiments with CD4-Cre, they confirm that Lis1 is less required for CD8 proliferation to TCR stimulation. However, it appears it is required in CD4 T cells. The authors show that the absence of Lis1 leads to p53 activation in CD4 T cells and DN3 thymocytes. They attempt to dissect the mechanism of the requirement for Li1 in CD4 T cells. Their results establish that CD4 T cells proliferating in the absence of Lis1 possess extra centrosomes.

The work here is well-performed and appropriately referenced. However, the puzzle that is not resolved, is why CD8 T cells have a lesser requirement for Lis1 after TCR stimulation. Because the CD4-Cre continues to be expressed and active in CD4 but not CD8 single-positive T cells, the authors should rigorously establish that the Lis1 protein is indeed ablated in CD8 T cells. i.e. they should establish that the proliferation of CD8 T cells is not due to escapees that continue to possess Lis1. Although this may appear unlikely because these cells do have proliferative defects in response to mitogens, this caveat should be rigorously excluded.

Additionally, the different responses in CD8 T cells could be due to a different proliferative mechanism, as the authors hypothesize, but also due to other differences. CD8 T cells may generate lower levels of p53, for example, or they may be resistant to the effects of p53. The authors should establish whether p53 stabilization is evident in CD8 T cells, as they see for DN3 and CD4 T cells. Additionally, they should determine whether extra centrosomes are seen in TCR-signaled CD8 T cells, as they show for CD4 T cells.

*Reviewer #3 (Recommendations for the authors):*

1. In Figure 1A it is clear that DN cells have increased frequencies in Lis1flox/flox-Cd2Cre mice. However, when numbers are calculated, there is no difference between control and LIS1-deficient mice. I understand that the composition of DN cells (DN1 to DN4) is different when comparing both groups, but this also suggests that there are fewer cells in the thymus of Lis1flox/flox-Cd2Cre animals. Did the authors observe faster thymic involution in these mice?

2. Can authors hypothesize what is shared between the maturation of pre-pro-B cells into pro-B cells and the transition from DN3 to DN4 in the thymus that could explain why LIS deficiency is particularly affecting these stages?

3. In Figures 1C/D authors claim that "LIS1 was not required for functional pre-TCR assembly but rather for the expansion of DN3 thymocytes…" based on the expression of IL-7R and CD5 by LIS-deficient DN3 thymocytes. However, authors previously state that "Notch and the IL-7receptor (IL-7R) stimulation lead to the up-regulation of CD5". As CD5 expression seems to be upregulated in DN3 thymocytes from Lis1flox/flox-Cd2Cre, it would be interesting to understand whether other signals downstream of Notch and IL-7R are being impacted. This would further strengthen the idea that signalling is not the reason for the defect in proliferation seen in these cells. This conclusion would also be strengthened by further evidence of proliferative defects in LIS1-deficient DN3 thymocytes – using Ki67 or BrdU staining combined with the DNA one. The experimental setup used to produce the data shown in Figures 2A and 2B could be used to address this question.

4. In Figure 2C the higher expression of CD5 by DN3 cells from Lis1flox/flox-Cd2Cre mice is not observed, as opposed to what is seen in Figure 1C. On the other hand, the phenotype of cells stuck in G2/M phase seems to be more severe (Figure 2E). Can authors discuss what could be the reason for these different phenotypes?

5. I understand that BCl^-^2 expression can be used as a measurement of cell survival, but the authors' claim that "the inability of cells to proliferate was not primarily due to survival defects" would be strengthened by direct measurement of cell viability.

6. Results depicted in Figure 3 (including its supplementary Figure) are mostly confirmations of a previous study (Ngoi, Lopez, Chang, Journal of Immunology, 2016; reference 34). Here the authors provide further evidence of a distinct role of LIS1 in CD4 T cell proliferation. However, the experimental setup chosen to show the physiological impact of the proliferation defect observed in LIS1-deficient CD4 T cells on immune responses is limited as it is restricted to a very early timepoint. The manuscript would benefit from data obtained in later time points following immunization: day 7/8 after immunization at peak of T cell expansion and >4 weeks after immunization when the T cell pool would be enriched for long-lived memory cells. Can Lis1flox/flox-Cd4Cre mice still form any memory pool?

7. Figure 4: MG-132, being a proteasome inhibitor, will have global effects on cell proteostasis, which itself can lead to defects in the cell cycle. Could authors confirm these results inducing metaphase arrest using a different strategy?

8. It is very interesting that the association between dynein and dynactin does not need to be stabilized in all cell division contexts. As observed in reference 34 and confirmed by the authors, CD8^+^ T cell proliferation following TCR engagement is not impacted by LIS1 deficiency. Does LIS1 have any homologue that could compensate for its loss in certain scenarios? Is the expression of LIS1 in wild-type cells changed over the course of stimulation/proliferation?

9. Authors discuss the possible role of LIS1 in maintaining symmetric cell divisions. However, at least in the context of CD8^+^ T cell proliferation/differentiation, the presence of ICAM-1 (and its engagement to LFA-1) has been shown to be a requirement for asymmetric cell division (one of the publications that explores that is referred by the authors: Chang et al., 2007, Science, reference 8). In the context of antigen presentation by APCs (such as in in vivo challenges), ICAM-1 is present, but this is not the case upon stimulation with anti-CD3 and anti-CD28 antibodies only, the method used by the authors in Figure 3. Thus, TCR engagement that most likely leads to symmetric divisions is being impaired in LIS1-deficient CD8^+^ T cells. Furthermore, even if authors assume that anti-CD3 and anti-CD28 only would promote polarization, CD4^+^ T cells have been shown to undergo asymmetric cell division (ref.8) after the first antigen encounter (reference 8). As the authors suggest that LIS1 is a requirement for symmetric cell divisions and not asymmetric ones, if no evidence of differences in asymmetric partitioning of cell cargoes upon loss of LIS1 is provided, I would suggest this session of the discussion be rephrased accordingly.

10. Did the authors ever check whether heterozygous mice (Cre+ but Lis1flox/+) have any phenotype? This would be interesting to provide preliminary insights into a point that is raised at the end of the discussion: "mono-allelic dysfunctions of LIS1might be sufficient to modulate the susceptibility to oncogenic transformation."

---

## [Author Response]

Reviewer #1 (Recommendations for the authors):a) It has been shown in the literature (e.g. https://doi.org/10.4049/jimmunol.1600827) that Cre itself can affect T cell differentiation. In this publication, the authors compared Cd2-Cre Lisflox/flox versus Lisflox/flox or Cd4-Cre Lisflox/flox versus Lisflox/flox, which means it is difficult to be sure that the effect is entirely due to the deletion of Lis1 and not partly due to the Cre itself. I think that it is important that they include this control.

We included new data showing that the Cre recombinase expression has no detectable effect on T-cell development and peripheral T-cell percentages in Lis1flox/+ mice when induced by the CD2 and the CD4 promoter respectively (Figure 1 —figure supplement 1B and Figure 3 —figure supplement 1E of the revised version).

b) In figure 1 it would be helpful to determine the effect of Lis1 deletion on icβTCR DN3a and DN3b cells, as opposed to DN3 cells, as the low levels of icβTCR at the DN3a stage may be obscuring the effect of Lis1.

To address the reviewer’s comment, we analyzed the intracytoplasmic expression of the TCRbeta chain versus the cell surface expression of CD27, a cell surface marker that discriminate DN3a (CD27^low^) from DN3b (CD27^hi^) thymocytes (Figure 1C of the revised manuscript). As expected, we show that the TCRbeta chain is mostly expressed in CD27^hi^ thymocytes. The percentages of TCRbeta^hi^CD27^hi^ thymocytes in DN3 cells were decreased in the absence of LIS1, suggesting a defect upon or after the beta-selection checkpoint. The absence of detectable signaling defect associated to the pre-TCR (Figure 1D), Notch (Figure 1E) and IL-7R (Figure 1E and 2E) suggest that LIS1 is important for cell expansion after the b-selection checkpoint.

c) In Cd4-Cre Lisflox/flox mice there was a drop in peripheral CD4^+^ and CD8^+^ cells and a 10% drop in proliferating CD8^+^ cells. The authors claim that this demonstrates that Lis1 is therefore not required for the proliferation of CD8 cells. However, as there is a slight effect, I do not feel that you can definitively claim that there is no effect. It would be useful to perform the cell cycle analysis and determine whether there is a defect in cell division, or not, to support this claim.

We agree with the reviewer and took out the statement that “LIS1 is not required for TCR-mediated proliferation” (page 8 of the previous manuscript) since it is true that a modest effect could be detected in Figure 3D. We believe that this effect is relatively marginal in comparison to the dramatic effect observed in CD4^+^ T cells after TCR engagement, where the numbers of cells that divided after the first round of division are decreased by 80 to 90%. Note that, we observed a nearly complete absence of proliferation of Lis1-deficient OTII+CD4^+^ T cell in vivo following immunization with ovalbumin, whereas the absence of LIS1 has no detectable effect on the expansion of OTI+CD8^+^ T cell in *Listeria*-Ova infection models (Ngoi, S. M. *et al.*, *The Journal of Immunology,* 2016). To address the reviewer’s comment, we analyzed the cell cycle of CD8^+^ T cells stimulated with anti-CD3 and anti-CD28 antibodies in similar conditions as for CD4^+^ T cells (Figure 3C). The loss of LIS1 did not impact on the percentages of cells in G2/M, suggesting further that CD8^+^ T cells divide normally in the absence of LIS1. We also include new data showing that p53 is expressed at similar level in wild-type and Lis1-deficient CD8^+^ T cells after stimulation (Figure 5B), indicating that the division of CD8^+^ T cells is not associated with p53-induced cell death in the absence of LIS1.

Also, I am unclear as to why there was a drop in peripheral CD8^+^ cells if Lis1 did not affect proliferation.

LIS1 is essential for IL-7-mediated homeostatic proliferation in CD8^+^ T cells (Ngoi, S. M. *et al.*, *The Journal of Immunology,* 2016) but seems more dispensable in this subset for TCR-mediated cell divisions. The reason of this is unclear but could result from distinct molecular requirements of cell divisions upon stimulation by soluble factors and antigenpresenting cells. This has been well characterized in previous studies from Krummel’s lab on Septins protein (Mujal, A.M. *et al.*, *Nature Immunology*, 2015). We show in Figure 3D that the loss of LIS1 leads to a strong defect of proliferation in CD8^+^ T cells when cell divisions are prompted by pharmacological activation, confirming that CD8^+^ T cells become sensitive to the loss of LIS1 when cell divisions are elicited by soluble factors.

d) In figure 4C, D you state that some cells are divided with an unequal repartition of the chromosomes in the daughter cells. Please state how this was measured.

The repartition of genetic material was scored based on the analysis of mitosis by video microscopy. Typical mitosis with equal chromosomes repartition (LIS1-WT CD4^+^ T cells) is represented in Video-1 of the revised manuscript whereas typical mitosis with unequal repartition is represented in Video-3. Mitosis with unequal repartition of DNA are typical in that they lead to the formation of several nuclei in one of the two daughter cells.

e) In some of the figure legends the authors state: Data represent one experiment out of two independent experiments with n=30-50 cells analyzed per group. Please clarify why you haven't shown the data from both experiments.

In Figure 6 A and C, the numbers of centrosomes per cell were quantified in two independent experiments. The percentages of cells with n=2 and n>2 centrosomes were calculated independently and were in similar ranges for both experiments. We decided not to combine those percentages on one single graph since it would lead to represent the average of only two values on which we could not really perform statistical tests. We also estimated that calculating a single value of percentage from two experiments would not be completely accurate since they represent independent analyses. In Figure 6B, the histogram bar graphs representing PCM size now include combined data from three independent experiments.

f) In the discussion there is a paragraph correlating the lack of effect on CD8 cells and asymmetric cell division. I think this is an interesting point and while I appreciate that is it possibly beyond the scope of this paper I do think that the authors should draw on the literature examining the occurrence of asymmetric cell division in CD4 cells versus CD8 cells to strengthen this section of the discussion.

A new chapter discussing this point is now included p18 in the revised version of the manuscript.

“This raises the question of whether CD4^+^ T cells would be more prone to symmetric divisions than CD8^+^ T cells. Theoretically, the experimental settings that we used in this study might not be optimal for eliciting asymmetric cell division, since we stimulated T cells with anti-CD3 and anti-CD28 in the absence of ICAM-1, which is required for asymmetric cell divisions to occur in the context of APC stimulation (Chang, J. T. et al., Science, 2007). However, the rate of asymmetric cell divisions might be less influenced by ICAM-1 stimulation in conditions where plate-bound stimulation with antibodies are used (Jung H.R. PLOS one, 2014). Asymmetric cell divisions have been detected in CD4^+^ T cells after the first antigen encounter (Chang, J. T. et al., Science, 2007), but it is unknown whether these divisions occur systematically or whether they occur with variable frequency that could be context-dependent. It is also unclear whether CD4^+^ and CD8^+^ T cells have similar rates of asymmetric division since the literature is lacking of quantitative studies in which cellular events associated to mitosis would be investigated side-by-side in those two subsets. The repartition of the transcription factor T-bet in daughter cells was compared in one study by flow cytometry in CD4^+^ and CD8^+^ T cells after a first round of cell division (Chang, J. T. et al., Immunity, 2011). Authors showed that T-bet segregates unequally in daughter cells in both CD4^+^ and CD8^+^ T cells. However, the disparity of T-bet between daughter cells was higher in CD8^+^ T cells as compared to that in CD4^+^ T cells (5- versus 3-fold), suggesting that cell-fate determinants are either more equally (or less unequally) distributed in daughter cells from the CD4^+^ lineage or that the rate of symmetric divisions is higher in CD4^+^ T cells than in the CD8^+^ T cells. More extensive analysis would be required to precisely quantify the rate of symmetric and asymmetric cell divisions in CD4^+^ and CD8^+^ T cells in the context of APC stimulation.”

Reviewer #2 (Recommendations for the authors):Lis1 is a dynein binding protein previously shown to be important for some types of T cell proliferation including homeostatic proliferation, but less important for the response of CD8 T cells to TCR stimulation. However, it is required for the development of CD8 T cell memory in model systems. In the present study, the authors repeat and extend the previous work. They construct iCD2-Cre mice harboring a conditional allele of Lis1 that it is critically required for proliferation in T cell development. Repeating earlier experiments with CD4-Cre, they confirm that Lis1 is less required for CD8 proliferation to TCR stimulation. However, it appears it is required in CD4 T cells. The authors show that the absence of Lis1 leads to p53 activation in CD4 T cells and DN3 thymocytes. They attempt to dissect the mechanism of the requirement for Li1 in CD4 T cells. Their results establish that CD4 T cells proliferating in the absence of Lis1 possess extra centrosomes.The work here is well-performed and appropriately referenced. However, the puzzle that is not resolved, is why CD8 T cells have a lesser requirement for Lis1 after TCR stimulation. Because the CD4-Cre continues to be expressed and active in CD4 but not CD8 single-positive T cells, the authors should rigorously establish that the Lis1 protein is indeed ablated in CD8 T cells. i.e. they should establish that the proliferation of CD8 T cells is not due to escapees that continue to possess Lis1. Although this may appear unlikely because these cells do have proliferative defects in response to mitogens, this caveat should be rigorously excluded.

To address this possibility, we verified that LIS1 was depleted similarly in both CD4^+^ and CD8^+^ peripheral T cells. Analysis by Western blot shows that LIS1 is efficiently depleted in both subsets, suggesting that the mild effect of LIS1 in CD8^+^ T cells is not resulting from the remaining expression of LIS1. Those data are now included as Figure 3 —figure supplement 2B of the revised version.

Additionally, the different responses in CD8 T cells could be due to a different proliferative mechanism, as the authors hypothesize, but also due to other differences. CD8 T cells may generate lower levels of p53, for example, or they may be resistant to the effects of p53. The authors should establish whether p53 stabilization is evident in CD8 T cells, as they see for DN3 and CD4 T cells. Additionally, they should determine whether extra centrosomes are seen in TCR-signaled CD8 T cells, as they show for CD4 T cells.

To address this comment, we analyzed the expression level of p53 in CD8^+^ T cells stimulated in similar conditions as for CD4^+^ T cells. In contrast to CD4^+^ T cells, we observed comparable amounts of p53 in wild-type and LIS1-deficient CD8^+^ T cells (Figure 5B), suggesting that the modest effect of LIS1 in CD8^+^ T cells is not resulting from a reduced p53 sensitivity threshold which would enable proliferation despite abnormal cell divisions. We also performed cell cycle analysis in CD8^+^ T cells and show that the loss of LIS1 does not affect the percentages of cells in the G2/M phase (Figure 3C).

Reviewer #3 (Recommendations for the authors):1. In Figure 1A it is clear that DN cells have increased frequencies in Lis1flox/flox-Cd2Cre mice. However, when numbers are calculated, there is no difference between control and LIS1-deficient mice. I understand that the composition of DN cells (DN1 to DN4) is different when comparing both groups, but this also suggests that there are fewer cells in the thymus of Lis1flox/flox-Cd2Cre animals. Did the authors observe faster thymic involution in these mice?

We did not perform this analysis since most of our phenotypic studies were done on mice that were younger than three months old, before the beginning of thymic involution in wild-type animals. We observed a dramatic decrease of thymus cellularity in young Cd2-Cre Lis1^flox/flox^- mice but we attributed this to the strong block of thymocyte proliferation after the bselection checkpoint. DN1 and DN2 populations were not decreased in numbers in the absence of LIS1 suggesting that not all thymocyte subsets are decreased such as during thymic involution processes. However, it is possible that the massive drop of thymocytes observed in the absence of LIS1 affect the thymic epithelial architecture and cause premature thymus involution.

2. Can authors hypothesize what is shared between the maturation of pre-pro-B cells into pro-B cells and the transition from DN3 to DN4 in the thymus that could explain why LIS deficiency is particularly affecting these stages?

The defects in T- and B- cell development observed in CD2-Cre Lis1^flox/flox^ mice occur at two different stages which are not directly related developmentally speaking. Within the T-cell lineage, the loss of LIS1 leads to a block after the b-selection checkpoint due to the inability of cells to efficiently proliferate. Within the B-cell lineage, LIS1 deficiency induces an early block prior the stage at which the pre-BCR is formed (pre-B cells), which theoretically could be compared to the DN2 stage in thymocytes. We did not investigate in detail the cause of the B-cell development defect but we could speculate that it might be the consequence of a proliferative block similar to what is observed in thymocytes. Pro-B cells were shown to proliferate in response to IL-7 stimulation (Hardy R.R *et al., The journal of experimental medicine*, 1991; Corfe S.A. *et al.*, *Seminars in Immunology*, 2012) and LIS1 is required for cell division upon IL-7 stimulation in peripheral CD8^+^ T cells (Ngoi, S. M. *et al.*, *The Journal of Immunology,* 2016). If this interpretation is correct, it is intriguing that we did not detect an effect of LIS1 at the DN2 stage of T-cell development, as DN2 thymocytes were also reported to proliferate in response to IL-7R signaling (von Freeden-Jeffry, U. et al., Immunity, 1997).

One possibility to explain this could be that the Lis1^flox/flox^ genes are deleted at a later stage of T-cell development than of B-cell development. Previous analyses of Cre recombinase activity in the hCD2-promoter model suggest that the Cre is effective in all B cell precursor subsets (Siegemund S. *et al., PLOS one*, 2015), whereas in thymocytes optimal recombination is reached at the DN3 stage (Shi J. *et al., PLOS one*, 2012).

3. In Figures 1C/D authors claim that "LIS1 was not required for functional pre-TCR assembly but rather for the expansion of DN3 thymocytes…" based on the expression of IL-7R and CD5 by LIS-deficient DN3 thymocytes. However, authors previously state that "Notch and the IL-7receptor (IL-7R) stimulation lead to the up-regulation of CD5". As CD5 expression seems to be upregulated in DN3 thymocytes from Lis1flox/flox-Cd2Cre, it would be interesting to understand whether other signals downstream of Notch and IL-7R are being impacted. This would further strengthen the idea that signalling is not the reason for the defect in proliferation seen in these cells. This conclusion would also be strengthened by further evidence of proliferative defects in LIS1-deficient DN3 thymocytes – using Ki67 or BrdU staining combined with the DNA one. The experimental setup used to produce the data shown in Figures 2A and 2B could be used to address this question.

As answered above, CD5 is a downstream target of the pre-TCR signaling but, to our knowledge, it is not a downstream target of Notch or IL-7R signaling. The sentence p7 of the manuscript was re-formulated since we understand that it could be misleading. However, we fully agree with the reviewer’s comment on Notch and IL-7R signaling and included new data in the revised version of the manuscript to address this point. Notch signaling stimulates metabolic changes which lead to the increase of thymocyte cell-size following the b-selection checkpoint (Ciofani M. *et al.*, *Nature Immunology*, 2005; Maillard I. *et al.*, *The Journal of Experimental Medicine*, 2006) and to the up-regulation of the transferrin receptor CD71 (Kelly, A.P. *et al.*, *The EMBO journal*, 2007). We now show in Figure 1E of the revised manuscript that the loss of LIS1 does not affect the average cell-size of post-b-selection thymocytes and the expression level of CD71 in these cells, suggesting that Notch signaling is preserved in the absence of LIS1. This was confirmed in vitro following stimulation of DN3a thymocytes with OP9-dl1 cells (Figure 2D of the revised manuscript). In this Figure, we also analyzed the expression level of BCl^-^2, which is regulated by IL-7R signaling (von Freeden-Jeffry, U. *et al.*, *Immunity*, 1997). We show that BCl^-^2 is comparable in abundance in LIS1 wild-type and LIS1deficient thymocytes following stimulation with OP-9dl1, suggesting that IL-7R signaling is not affected by the absence of LIS1. As for proliferative defects, we performed experiments to stain KI67 in thymocytes subsets but failed to detect any differences in WT and LIS1 deficient DN3 thymocytes (Author response image 1). However, the flow cytometry profile of KI67 expression in WT cells was almost similar in DN2, DN3-CD5^low^ and DN3-CD5^hi^ subsets, three populations which should exhibit different percentages of proliferative cells. The dot plot analysis of CD5 versus KI67 staining in DN3 thymocytes shows no detectable increase of KI67 expression in CD5hi thymocytes which is unexpected since these cells are enriched in post-b-selection thymocytes which extensively proliferate. A small decrease of Ki67 MFI was detected in DN3-CD5^low^ thymocytes as compared to that in other subsets but it seems to correspond to a general shift of the staining rather than a decrease of a KI67^hi^ subset. We decided not to include this analysis since we suspect that our staining did not efficiently discriminate proliferative events in these subsets.

**Author response image 1. sa2fig1:** KI67 expression profile in thymocytes from Lis1^flox/flox^ and CD2-Cre Lis1^flox/flox^ mice. Left panel. Histogram overlay represent KI67 staining on DN3-CD5^hi^ thymocytes from Lis1^flox/flox^ and CD2-Cre Lis1^flox/flox^ mice. Middle panel. Histograms represent KI67 staining on the indicated thymocyte subsets from Lis1^flox/flox^ (WT) mice. Right panel. Dot plot represent CD5 versus Ki67 staining on DN3 thymocytes.

4. In Figure 2C the higher expression of CD5 by DN3 cells from Lis1flox/flox-Cd2Cre mice is not observed, as opposed to what is seen in Figure 1C. On the other hand, the phenotype of cells stuck in G2/M phase seems to be more severe (Figure 2E). Can authors discuss what could be the reason for these different phenotypes?

For the cell cycle analysis, the effects of LIS1 are relatively similar in vivo and in vitro if you consider the mean values represented in the bar graphs. More representative histograms are now included in Figure 2F of the revised version. For CD5 expression, we suspect that the higher expression of CD5 on LIS1 deficient DN3 thymocytes is not the consequence of a direct effect of LIS1 on pre-TCR signaling, as suggested by the absence of effect on CD5 when cells are stimulated in vitro with OP-9dl1 cells, but an indirect effect due to the accumulation DN3 thymocytes that are blocked at this stage of T-cell development. CD5 surface expression is stimulated by pre-TCR signaling and CD5 increases in abundance as cells progress from the DN3 stage to the DN4 and DP stage (Azzam, H. S. *et al., The journal of experimental medicine,* 1998). We think that part of the cells characterized as DN3 cells in LIS1 deficient mice represent cells that have failed to transit to the DN4 stage and have accumulated pre-TCR signals, leading to the higher surface expression of CD5. This is not observed in vitro presumably because DN3 cells received synchronized stimuli by OP9 cells for relatively short periods of time.

5. I understand that BCl^-^2 expression can be used as a measurement of cell survival, but the authors' claim that "the inability of cells to proliferate was not primarily due to survival defects" would be strengthened by direct measurement of cell viability.

The effect of LIS1 on cell viability is indirectly addressed in Figure 5, for both thymocytes and peripheral CD4^+^ T cells, by evaluating the percentages of apoptosis in divided and undivided cells. Similar results as the one represented in Figure 5 were obtained at day 3 after stimulation, when viability dyes were used instead of annexin-5. This was performed in only one experiment and was not included in the manuscript. We agree with the reviewer that the sentence “The loss of LIS1 also did not affect the expression of BCl^-^2 (Figure 2D), …, suggesting that the inability of cells to proliferate was not primarily due to survival defects” appeals for a following-up analysis of cell survival in the same figure. We changed this sentence and presented the analysis of BCl^-^2 expression as a read-out of IL-7R signaling instead of cell survival (p8).

6. Results depicted in Figure 3 (including its supplementary Figure) are mostly confirmations of a previous study (Ngoi, Lopez, Chang, Journal of Immunology, 2016; reference 34). Here the authors provide further evidence of a distinct role of LIS1 in CD4 T cell proliferation. However, the experimental setup chosen to show the physiological impact of the proliferation defect observed in LIS1-deficient CD4 T cells on immune responses is limited as it is restricted to a very early timepoint. The manuscript would benefit from data obtained in later time points following immunization: day 7/8 after immunization at peak of T cell expansion and >4 weeks after immunization when the T cell pool would be enriched for long-lived memory cells. Can Lis1flox/flox-Cd4Cre mice still form any memory pool?

We choose to analyze CD4^+^ T cells at day 2 and 3 after immunization because we sought to catch early cell-division waves through CTV dilution. We also wanted to show that LIS1 deficient CD4^+^ T cells could normally survive and migrate to lymph nodes before they start to proliferate. Given the dramatic effect of LIS1 on CD4^+^ T-cell proliferation at day 3, we anticipated that very low numbers of LIS1 deficient cells would survive at later time points after immunization. To address the reviewer’s comment, we transferred OT2+CD45.1+ CD4^+^ T cells stained with CTV in C57BL/6 mice and analyzed the percentages and numbers of CD45.1+ T cells as well as the dilution of CTV in those cells at day 7 after immunization. As expected, all CD45.1+ cells were negative for CTV within this time of analysis (data not shown). The percentages and numbers of CD45.1+ T cells were strongly decrease in the absence of LIS1 in comparison to wild-type controls (Figure 3 —figure supplement 2C), confirming results obtained at day 3 after immunization. We think that it would be very difficult to infer possible effects of LIS1 on long-lived memory cells given the bias introduced by the major proliferative defect observed at early times.

7. Figure 4: MG-132, being a proteasome inhibitor, will have global effects on cell proteostasis, which itself can lead to defects in the cell cycle. Could authors confirm these results inducing metaphase arrest using a different strategy?

We are not aware of an alternative to MG132 treatment, which is a very classic approach to our knowledge to induce metaphase arrest. We believe that the inability of LIS1deficient cells to rapidly reach metaphasis was confirmed in the video microscopy studies (Figure 4C and video-2 and -3) in which cells were not incubated with MG132.

8. It is very interesting that the association between dynein and dynactin does not need to be stabilized in all cell division contexts. As observed in reference 34 and confirmed by the authors, CD8^+^ T cell proliferation following TCR engagement is not impacted by LIS1 deficiency. Does LIS1 have any homologue that could compensate for its loss in certain scenarios? Is the expression of LIS1 in wild-type cells changed over the course of stimulation/proliferation?

We agree that this is a puzzling observation and can only speculate at this stage for an explanation. We are not aware of any LIS1 paralogs or other molecules that could substitute for LIS1 deficiency. The abundance of LIS1 remains unchanged in CD8^+^ T cells at day two after stimulation with anti-CD3 and anti-CD28 antibodies, when cell divisions have been initiated (data not shown). Of note, the interaction between dynein and the dynactin component p150Glued was not totally disrupted in CD4^+^ T cells lacking LIS1, suggesting that dynein and dynactin interact to some extent through LIS1-independent mechanism. A possibility to explain this observation would be that LIS1 acts in a context-dependent manner on specific dynein/dynactin complexes which would be more critical for the division of CD4^+^ T cells than of CD8^+^ T cells. Dynein assembles with many other regulatory proteins which lead to the formation of different dynein complexes, each serving different purposes upon cell division (kinetochore localization, force generation at the cell cortex, chromosomes alignment, etc…) (Raaijmakers J.A. *et al.*, *The journal of Cell biology*, 2013; Torisawa T. *et al.*, *Frontiers in Cell and Developmental Biology*, 2020). Although LIS1 is associated with the majority of those complexes (Raaijmakers J.A. *et al.*, The journal of Cell biology, 2013), it is not the case for dynactin components which associate in a restrictive manner to some dynein complexes but not to others (Raaijmakers J.A. *et al.*, *The journal of Cell biology*, 2013). Previous studies have shown context-dependent effect of LIS1 (Baumbach J. *et al.*, *eLife, 2017*), suggesting that LIS1 might regulate dynein/dynactin interaction in some complexes but not in others. Why those complexes would be particularly important for the division of CD4^+^ T cells but not of CD8^+^ T cells would need further investigations, but could possibly be related to differences linked to the symmetry of cell division, which implies different mechanistic readjustments of mitotic spindles presumably involving distinct dynein/dynactin-dependent processes.

9. Authors discuss the possible role of LIS1 in maintaining symmetric cell divisions. However, at least in the context of CD8^+^ T cell proliferation/differentiation, the presence of ICAM-1 (and its engagement to LFA-1) has been shown to be a requirement for asymmetric cell division (one of the publications that explores that is referred by the authors: Chang et al., 2007, Science, reference 8). In the context of antigen presentation by APCs (such as in in vivo challenges), ICAM-1 is present, but this is not the case upon stimulation with anti-CD3 and anti-CD28 antibodies only, the method used by the authors in Figure 3. Thus, TCR engagement that most likely leads to symmetric divisions is being impaired in LIS1-deficient CD8^+^ T cells. Furthermore, even if authors assume that anti-CD3 and anti-CD28 only would promote polarization, CD4^+^ T cells have been shown to undergo asymmetric cell division (ref.8) after the first antigen encounter (reference 8). As the authors suggest that LIS1 is a requirement for symmetric cell divisions and not asymmetric ones, if no evidence of differences in asymmetric partitioning of cell cargoes upon loss of LIS1 is provided, I would suggest this session of the discussion be rephrased accordingly.

We agree with the reviewer and have rephrased the discussion as followed (p18, of the manuscript):

“This raises the question of whether CD4^+^ T cells would be more prone to symmetric divisions than CD8^+^ T cells. Theoretically, the experimental settings that we used in this study might not be optimal for eliciting asymmetric cell division, since we stimulated T cells with anti-CD3 and anti-CD28 in the absence of ICAM-1, which is required for asymmetric cell divisions to occur in the context of APC stimulation (Chang, J. T. et al., Science, 2007). However, the rate of asymmetric cell divisions might be less influenced by ICAM-1 stimulation in conditions where plate-bound stimulations with antibodies are used (Jung H.R. PLOS one, 2014). Asymmetric cell divisions have been detected in CD4^+^ T cells after the first antigen encounter (Chang, J. T. et al., Science, 2007), but it is unknown whether these divisions occur systematically or whether they occur with variable frequencies that could be context dependent. It is also unclear whether CD4^+^ and CD8^+^ T cells have similar rates of asymmetric division since the literature is lacking of quantitative studies in which cellular events associated to mitosis would be investigated side-by-side in those two subsets. The repartition of the transcription factor T-bet in daughter cells was compared in one study by flow cytometry in CD4^+^ and CD8^+^ T cells after a first round of cell division (Chang, J. T. et al., Immunity, 2011). Authors showed that T-bet segregates unequally in daughter cells in both CD4^+^ and CD8^+^ T cells. However, the disparity of T-bet between daughter cells was higher in CD8^+^ T cells as compared to that in CD4^+^ T cells (5- versus 3-fold), suggesting that cell-fate determinants are either more equally (or less unequally) distributed in daughter cells from the CD4^+^ lineage or that the rate of symmetric divisions is higher in CD4^+^ T cells than in the CD8^+^ T cells. More extensive analysis would be required to precisely quantify the rate of symmetric and asymmetric cell divisions in CD4^+^ and CD8^+^ T cells in the context of APC stimulation.”

10. Did the authors ever check whether heterozygous mice (Cre+ but Lis1flox/+) have any phenotype? This would be interesting to provide preliminary insights into a point that is raised at the end of the discussion: "mono-allelic dysfunctions of LIS1might be sufficient to modulate the susceptibility to oncogenic transformation."

We have addressed this point experimentally in the revised version of the manuscript. We show that mono-allelic LIS1 deficiency does not have a significant impact on the percentages of thymocyte populations in Cd2-Cre Lis1flox/+ mice (Figure 1 —figure supplement 1B) and on the numbers of peripheral T cells in Cd4-Cre Lis1flox/+ (Figure 3 —figure supplement 1E), suggesting that LIS1 does not operate in a dose-dependent fashion in the context of T-cell development and T-cell homeostatic maintenance. Additionally, Cd4Cre Lis1flox/+ CD4^+^ T cells proliferate effectively following TCR and CD28 stimulation (Figure 3 —figure supplement 2A), indicating further that mono-allelic LIS1 dosage is sufficient to support cell division of CD4^+^ T cells. The part of the discussion related to Lis1 haplo-deficiency has been rephrased according to this new set of data.